

# Three-dimensional data-tracking simulations of sprinting using a direct collocation optimal control approach

Nicos Haralabidis[1,2], Gil Serrancolí[3], Steffi Colyer[1,2], Ian Bezodis[4], Aki Salo[1,2,5] and Dario Cazzola[1,2]

[1] Department for Health, University of Bath, Bath, UK
[2] CAMERA-Centre for the Analysis of Motion, Entertainment Research and Applications, Bath, UK
[3] Department of Mechanical Engineering, Universitat Politècnica de Catalunya, Barcelona, Spain
[4] Cardiff School of Sport and Health Sciences, Cardiff Metropolitan University, Cardiff, UK
[5] KIHU Research Institute for Olympic Sports, Jyväskylä, Finland

Corresponding author
Nicos Haralabidis,
nh546@bath.ac.uk

## ABSTRACT

Biomechanical simulation and modelling approaches have the possibility to make a meaningful impact within applied sports settings, such as sprinting. However, for this to be realised, such approaches must first undergo a thorough quantitative evaluation against experimental data. We developed a musculoskeletal modelling and simulation framework for sprinting, with the objective to evaluate its ability to reproduce experimental kinematics and kinetics data for different sprinting phases. This was achieved by performing a series of data-tracking calibration (individual and simultaneous) and validation simulations, that also featured the generation of dynamically consistent simulated outputs and the determination of foot-ground contact model parameters. The simulated values from the calibration simulations were found to be in close agreement with the corresponding experimental data, particularly for the kinematics (average root mean squared differences (RMSDs) less than 1.0° and 0.2 cm for the rotational and translational kinematics, respectively) and ground reaction force (highest average percentage RMSD of 8.1%). Minimal differences in tracking performance were observed when concurrently determining the foot-ground contact model parameters from each of the individual or simultaneous calibration simulations. The validation simulation yielded results that were comparable (RMSDs less than 1.0° and 0.3 cm for the rotational and translational kinematics, respectively) to those obtained from the calibration simulations. This study demonstrated the suitability of the proposed framework for performing future predictive simulations of sprinting, and gives confidence in its use to assess the cause-effect relationships of technique modification in relation to performance. Furthermore, this is the first study to provide dynamically consistent three-dimensional muscle-driven simulations of sprinting across different phases.

## INTRODUCTION

Sprinting is the fastest mode of human bipedal locomotion, and the short distance events (60–400-m) in modern athletics provide a means of assessing the limits of human sprinting performance. The objective for athletes competing within these events is to cover the set distance in the shortest possible time, and often the winning margin is hundredths of a second at the highest levels of competition. Coaches and athletes are therefore continually striving for improvements in technique which can enhance overall performance by such fine margins. Sprinting also plays an important role within team-based sports. For instance, sprinting has been shown to be pivotal in creating goal scoring opportunities within soccer (*Faude, Koch & Meyer, 2012*). Thus, the insights from further understanding sprinting technique and performance can have far reaching applications, as they can also be transferred to benefit performance in other sports.

Most studies to date have advanced the knowledge of how to improve sprinting performance by assessing the ground reaction force (GRF) (*Colyer, Nagahara & Salo, 2018*; *Morin, Edouard & Samozino, 2011*), joint kinematics and kinetics (*Bezodis, Kerwin & Salo, 2008*; *Schache et al., 2019*; *Smith, Lake & Lees, 2014*), and spatiotemporal parameters (*Hunter, Marshall & McNair, 2004*; *Salo et al., 2011*). The studies focusing on joint kinetics (net joint moments, power, and work) potentially bear the most significance, as they can explain the causes of motion. However, the net joint moments calculated from inverse dynamics analyses (IDA) are known to be impacted by several factors (*Derrick et al., 2020*) (e.g. filtering and soft-tissue artefact), and necessitate fictitious residual forces and moments to be applied at the model's root segment (e.g. pelvis) due to dynamic inconsistencies. These residuals are further exacerbated during explosive tasks, such as sprinting, where experimental errors and modelling assumptions are likely to become more critical. Methods such as the residual reduction algorithm (RRA) within OpenSim (*Delp et al., 2007*) and optimal control approaches (*Lin & Pandy, 2017*; *Meyer et al., 2016*; *Pallarès-López et al., 2019*) have been introduced to compensate for the residuals, although they have not seen widespread adoption within the sports biomechanics literature. Furthermore, within the sports biomechanics literature these residuals are typically neglected, raising questions on the errors in IDA and the validity of the corresponding findings.

A further limitation within the current body of sprinting literature is that most studies have focused on identifying key aspects of technique from group level analyses. It is possible that the technique based findings from group level analyses can be transferred to benefit the performance levels of elite athletes. However, individualised technique modifications are more likely to be necessary for elite athletes to further improve their performance levels by the sought-after small margins. Furthermore, the existing literature provides various technique related factors associated with improved sprinting performance, as opposed to how specific technique modifications influence sprinting performance. Predictive computer simulation and modelling approaches within sports biomechanics can be used to overcome the aforementioned limitations, as they can be used

to identify optimum technique on an individualised basis, explore cause-effect relationships and assess 'what-if' scenarios (*Neptune, 2000*).

Recently, there has been a noticeable increase in the number of modelling and simulation studies within sports biomechanics adopting an optimal control theory approach (*Jansen & McPhee, 2020*; *Lin & Pandy, 2017*; *Porsa, Lin & Pandy, 2016*), also specifically within sprinting (*Celik & Piazza, 2013*; *Schultz & Mombaur, 2010*). *Celik & Piazza (2013)* found that their optimised solution of sprinting demonstrated several salient features that have been observed experimentally (e.g. forward trunk lean during early acceleration), although their model lacked a sufficient representation of the lower-limbs to adequately characterise their actions. Conversely, *Schultz & Mombaur (2010)* developed a three-dimensional model, and used it in an exploratory sense to determine the feasibility of performing predictive sprinting simulations. However, a limitation common to both of these studies is that they did not explicitly evaluate their outputs against experimental data.

The evaluation of a modelling and simulation framework is needed to ensure that it is fit for its purposes and produces realistic results (*Yeadon & Challis, 1994*). Furthermore, within applied sports contexts the evaluation step is necessary to ensure that the results from predictive simulations can be transferred with confidence to drive real world changes. To conduct the evaluation step, data-tracking simulations using optimal control theory can be performed (*Umberger & Miller, 2017*), as they provide the opportunity to quantitatively determine how a modelling and simulation framework performs at reproducing experimental data. In addition, a major benefit of optimal control based data-tracking simulations is that they provide an elegant approach to improve the dynamic consistency of a simulation by either minimising or constraining the residuals, particularly when used with the direct collocation method (*Lin & Pandy, 2017*; *Meyer et al., 2016*; *Pallarès-López et al., 2019*). To date, however, no study has attempted to use this approach within a highly demanding sporting movement, such as sprinting.

Lastly, predictive simulations of sprinting necessitate a means of modelling the foot-ground interaction. One popular approach relies on distributing a finite set of contact elements across the surface of a foot and using a compliant foot-ground contact model to calculate the contact forces at each element (*Dorschky et al., 2019*; *Falisse et al., 2019a, 2019b*; *Gilchrist & Winter, 1996*; *Lin & Pandy, 2017*; *Serrancolí et al., 2019*). The difficulty with using this approach is determining the foot-ground contact model parameters. Previous studies have obtained the foot-ground contact model parameters by fitting their model to mechanical testing data (*Dorschky et al., 2019*; *Miller & Hamill, 2015*), thus the values of the parameters can be used by further studies. However, the use of previously published foot-ground contact model parameter values can be problematic due to the differences in the formulations and constitutive laws between compliant foot-ground contact models. It is also possible to determine the values of the foot-ground contact model parameters by manually tuning them (*Gilchrist & Winter, 1996*), although this approach can be challenging to implement due to the nonlinear interactions of the parameters. An alternative approach is to perform data-tracking

simulations, as they enable the foot-ground contact model parameter values to be determined in addition to the evaluation of a modelling and simulation framework (*Falisse et al., 2019a*; *Serrancolí et al., 2019*). When using the latter approach, care must be taken to ensure that the parameters obtained are not unrealistic and overfit, particularly if they are determined from tracking a single trial. Consequently, further independent data-tracking simulations using the obtained foot-ground contact model parameters must be performed to ensure that they lead to simulated outputs that closely resemble the experimental data, and to provide confidence for their use within predictive simulations.

The main aims of this study were to develop a musculoskeletal modelling and simulation framework for sprinting, and to evaluate its capability of reproducing experimental data (GRF, kinematics, kinetics and electromyograms–EMGs) by performing data-tracking simulations. The secondary aims of this study were to improve the dynamic consistency of the simulated outputs by enforcing the residuals to be zero, thus further highlighting the advantages of the approach used, and to identify foot-ground contact model parameters such that they can be used within future predictive simulations of sprinting.

## MATERIALS AND METHODS

### Experimental data collection

An international-level male sprinter (age: 24 years; height: 1.79 m; mass: 72.2 kg; 100 m PB: 10.33 s; 200 m PB: 20.27 s) provided written informed consent to participate in the current study. Ethical approval for this study was obtained from the University of Bath's Research Ethics Approval Committee for Health (EP 17/18 238). The data collection took place at the National Indoor Athletics Centre in Cardiff, UK (Fig. 1). The protocol required the athlete to perform a series of maximal sprinting trials whilst GRF, three-dimensional marker trajectories and EMGs were recorded.

Five force plates (×4 type: 9287CA and ×1 type 9282BA, Kistler, Switzerland) were used to collect GRF at a sampling frequency of 2,000 Hz. Three-dimensional marker trajectories were recorded using a 15-camera motion capture system (Qualisys AB, Sweden) sampling at 250 Hz. The motion capture system was positioned on either side of the track segment which enclosed the force plates. Forty-nine retro-reflective markers were attached to the surface of the athlete's skin and shoes using double-sided adhesive and medical tapes (Fig. 1). Eight acrylic marker clusters were also attached to the athlete using double-sided adhesive tape and medical bandages. Six wireless surface electrodes (Trigno Avanti, Delsys Inc., Boston, MA, USA) sampling at 2,000 Hz with a 20–450 Hz bandwidth were used to record EMGs from the following right lower-limb muscles of the athlete: gluteus maximus (GMAX), biceps femoris (BF), medial gastrocnemius (GASTM), tensor fasciae latae (TFL), vastus medialis (VM) and soleus (SOL). The skin preparation and placement of the electrodes was in accordance with previously published guidelines (*Hermens et al., 2000*).

The data collection commenced with a standing static trial of the athlete being captured. The athlete then performed two successful maximal effort sprints over three distances (0–10, 0–30 and 0–60 m) following the completion of a self-led warm up. The chosen

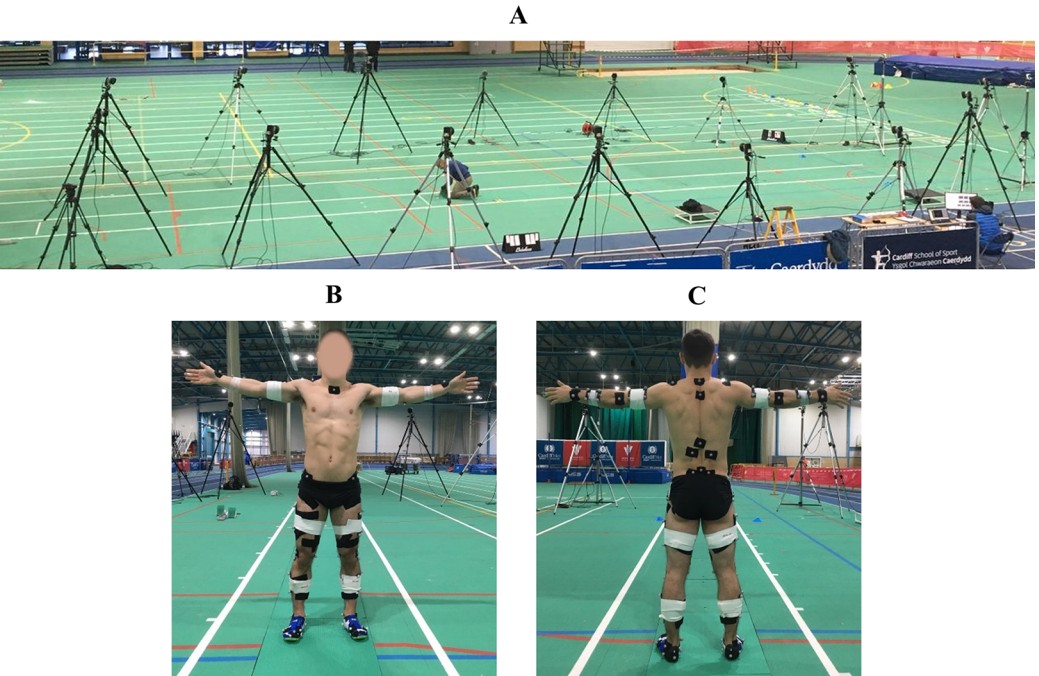

**Figure 1 Experimental data collection setup (A), and placement of retro-reflective markers, acrylic clusters, and EMG electrodes (B and C).** The force plates setup provided a 4.5 m segment of the track to capture GRF, and this permitted GRF data from a minimum of two successive steps to be captured for each trial. The calibrated motion capture volume covered approximately 9.5 (length) × 1.5 (width) × 2 (height) m³.                                               

distances covered the three conventional phases (early acceleration, mid-acceleration and maximum velocity) of short distance sprinting events. Different starting positions, relative to the first force plate, were used to enable the sprinter to contact the force plates between 0–5, 10–15 and 45–50 m for each of the distances. A trial was deemed successful if the athlete's entire foot landed within the boundaries of a single force plate whilst not noticeably altering their step. A rest period of up to 10 minutes between each sprint was given to the athlete. Each sprint was initiated by standard 'on your marks' and 'set' commands by a member of the research team. The researcher then pressed a trigger button that provided an auditory sound signal through a sounder device (Wee Beastie Ltd, UK), and this also triggered the synchronous acquisition of GRF, three-dimensional marker trajectories and EMGs through Qualisys Track Manager (version 2018.1; Qualisys AB, Sweden).

## Model

We started with a generic three-dimensional full-body musculoskeletal OpenSim model (*Hamner, Seth & Delp, 2010*) (Fig. 2) that had been previously used for applications in sprinting (*Dorn, Schache & Pandy, 2012*; *Lai et al., 2016*). The original model represented the human skeleton as a multibody system comprising 20 rigid segments and 29 degrees-of-freedom (DOFs). We added DOFs for subtalar inversion-eversion, metatarsophalangeal (MTP) dorsiflexion-plantarflexion, and wrist flexion-extension and

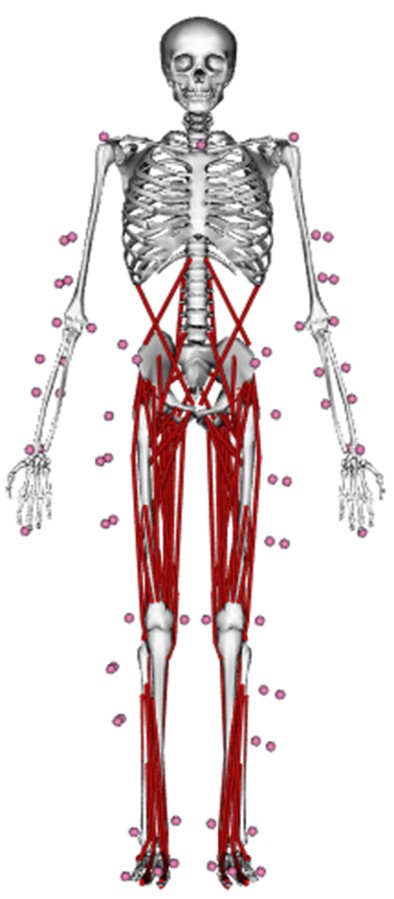

**Figure 2** **Three-dimensional musculoskeletal model (*Hamner, Seth & Delp, 2010*) used in the current study.** The same model was used to determine experimental kinematics and kinetics from inverse kinematics and dynamics analyses, respectively, and perform data-tracking simulations. The virtual model markers are denoted by pink spheres.   

adduction-abduction. The configuration of the modified model was uniquely described by a set of generalised coordinates $q \in \mathbb{R}^{37}$, with each DOF being represented by a generalised coordinate. The modified model's 37 DOFs were as follows: six DOFs ground-to-pelvis joint, three DOFs hip joints, one DOF knee joints, one DOF ankle joints, one DOF subtalar joints, one DOF MTP joints, three DOFs back joint, three DOFs shoulder joints, two DOFs elbow joints and two DOFs wrist joints.

The generic modified model was linearly scaled to match the anthropometric and inertial characteristics of the athlete by using a measurement-based approach within OpenSim. Scaling factors for each segment were calculated using the relative distances between markers placed over anatomical landmarks recorded in the static trial and corresponding virtual markers which were placed on the modified generic model. A combination of uniform and non-uniform scaling factors were used to scale the properties of each segment, and the lower- and upper-limbs were scaled to maintain bilateral symmetry.

The upper-limbs were actuated by 14 joint actuators whose limits were set based upon the IDA results. The lower-limbs and trunk were actuated by 92 Hill-type muscle-tendon

units (MTUs), and we used the dimensionless characteristic equations presented by *De Groote et al. (2016)* to describe the active and passive (series and parallel) components of force generation for each MTU. Five parameters are used to scale the dimensionless characteristic equations of Hill-type models for a specific MTU: pennation angle at optimum fibre length, optimum fibre length, tendon slack length, maximum shortening velocity and maximum isometric force. In our case we used the pennation angle at optimum fibre length parameters reported in the generic model, whilst we used the optimum fibre length and tendon slack length parameters obtained following the scaling of our model. The scaling procedure updated those parameters while preserving the ratio between them from the generic model. We set the maximum shortening velocity parameter of all the MTUs to be 12 optimal fibre lengths per second based upon an in vivo estimate of the maximum shortening velocity of human muscle (*De Ruiter et al., 2000*). The maximum isometric force parameters were set to be two times greater than those reported within the generic model, and this enabled the MTUs of the lower-limbs DOFs, except the MTP DOFs, to exert net moments that more closely reflected the training status of our athlete. For example, the four MTUs responsible for extension of the knee DOF had the capacity to exert an isometric net moment that was approximately 18% greater than reported by *Erskine et al. (2011)* for a group of untrained individuals following 9 weeks of resistance training. We also included reserve actuators for each of the lower-limb and trunk DOFs. The upper limits of the reserve actuators for the MTP DOFs and the remaining lower-limb and trunk DOFs were set to 40 and 10 Nm, respectively. The reserve actuators for the MTP DOFs were set with a higher upper limit such that together with the four MTUs spanning each of the MTP DOFs they had the capacity to produce net MTP moments that were in a similar range to those reported from isometric dynamometry testing (*Goldmann et al., 2013*). To account for the behaviour of the athlete's sprinting spikes and the passive structures surrounding the MTP DOFs we included a simple linear rotational spring. The stiffness of the rotational spring was set to 65 Nm/rad by combining the rotational stiffness of sprinting spikes/sports shoes (*Oh & Park, 2017*; *Toon et al., 2006*) with the rotational stiffness used previously for representing the passive structures surrounding the MTP DOFs (*Sasaki, Neptune & Kautz, 2009*).

Polynomials were used to describe the lengths, velocities and moment arms of the MTUs as functions of the model's generalised coordinates and velocities (*Falisse et al., 2019a*). The coefficients of the polynomials were determined by fitting them to the lengths and moment arms of the scaled model's MTUs, which were obtained by performing a Muscle Analysis within OpenSim for a wide range of generalised coordinates values. We opted to describe the lengths, velocities and moment arms of the MTUs with differentiable and continuous polynomial functions as they are ideally suited to the gradient-based optimal control approach we used.

The aerodynamic drag force within sprinting is known to have an influence on overall performance (*Quinn, 2003*), even when analysing performance across multiple steps with zero wind velocity (*Colyer, Nagahara & Salo, 2018*). To account for the aerodynamic drag force we used the approach presented by *Samozino et al. (2016)*, and

for this study we assumed that the wind velocity was zero as the experimental data collection took place in an indoor athletics centre. The aerodynamic drag force was applied to the pelvis segment, specifically at the model's centre of mass position expressed in the local frame of the pelvis segment. We opted to apply the aerodynamic drag force to the pelvis segment as this segment was the closest to the model's centre of mass position when visualised within the OpenSim graphical user interface for the trials processed.

## Data processing and analysis

An open-source MATLAB toolbox (https://simtk.org/projects/matlab_tools) was used to convert the raw data into the file formats compatible with OpenSim (version 3.3; Stanford University, CA, USA) (*Delp et al., 2007*). For the purposes of this study we used experimental data that spanned an arbitrary right foot stance phase alongside portions of the flight phases prior to touchdown and after take-off from the first early acceleration and mid-acceleration phase trials and from both the maximum velocity phase trials. Touchdown and take-off were determined using a 20 N vertical GRF threshold. The analysed flight phases were identified by searching within the vertical GRF signal 50 frames backwards and forwards from touchdown and take-off, respectively, to then identify the frames which most closely coincided with the sampling intervals of the motion capture system.

Global pelvis and relative joint kinematics for each trial were determined by performing inverse kinematics analyses within OpenSim from the recorded marker trajectories. The kinematics and GRF were filtered using a common 20 Hz fourth-order low-pass Butterworth filter. The cut-off frequency was determined by performing a residual analysis on the kinematics obtained from the inverse kinematics analyses (*Winter, 2009*). The filtered kinematics were also fitted using B-splines to enable velocities and accelerations to be determined. Net joint moments and pelvis residuals were calculated for each trial by performing IDA using the OpenSim C++ application programming interface. The splined kinematics, net joint moments and filtered GRF served as the experimental data to be tracked within the data-tracking simulations.

The EMGs from each trial were full-wave rectified and filtered at 20 Hz using a fourth-order low-pass Butterworth filter. We selected the cut-off frequency based on previous studies which assessed EMGs during running and sprinting (*Santuz et al., 2020*; *Yong, Silder & Delp, 2014*). Each filtered EMG was normalised to the maximum filtered amplitude of the respective EMG from within all the trials processed. Following the collection of the mid-acceleration and maximum velocity phase trials we noticed that the TFL and SOL electrodes, respectively, became detached from the surface of the athlete's skin, consequently we excluded the TFL and SOL EMG from those trials during data processing and further analysis.

## Optimal control problem formulation

The data-tracking simulations (calibration and validation) were formulated as optimal control problems (OCPs). The objective of the data-tracking simulations was to determine the state $x$ and control $u$ variables (plus the foot-ground contact model parameters $p$
for the calibration simulations), that resulted in tracking the experimental data as closely as possible, while satisfying our model's constraints.

The skeletal dynamics of our musculoskeletal model (Fig. 2) were described by a collection of 37 coupled second-order nonlinear differential equations. The contraction and activation dynamics of each MTU were described by two first-order nonlinear differential equations (*De Groote et al., 2016*; *Zajac, 1989*). Four state variables ($x = [q\ v\ F_T\ a]$, where $q$ and $v$ are the scaled musculoskeletal model's generalised coordinates and velocities, respectively, and $F_T$ and $a$ are the MTU normalised tendon forces and activations, respectively) were selected to enable the skeletal, contraction and activation dynamics to be represented as a system of first-order differential equations. We also included control variables for the upper-limb joint actuators $u_{UL}$ and the reserve actuators $u_{RES}$ that defined the instantaneous moment those actuators could produce.

We formulated the differential equations governing skeletal and contraction dynamics implicitly (*Falisse et al., 2019a*) by introducing additional control variables for the time derivatives of the generalised velocities $u_{\dot{v}}$ and normalised tendon forces $u_{\dot{F}_T}$. The implicit formulation led to enforcing the first-order skeletal and contraction dynamics with simple constraints, and additional nonlinear equality path constraints were used to ensure the skeletal and contraction dynamics equations were enforced. Two sets of equality path constraints were used to enforce the skeletal dynamics equations of our model's 37 DOFs. The first set enforced that the pelvis residuals (moments and forces at the six pelvis DOFs) obtained from evaluating the skeletal dynamics equations were zero. The second set enforced that the net joint moments (moments at the 31 relative DOFs) obtained from evaluating the skeletal dynamics equations matched with those obtained from the actuators. The inclusion of the control variables for the upper-limb actuators was not mandatory since we used idealised upper-limb actuators together with an implicit skeletal dynamics formulation. For the contraction dynamics equations we imposed equality path constraints to enforce the Hill-equilibrium condition (the normalised tendon force matched the projected normalised muscle force).

The differential equation describing activation dynamics was also formulated implicitly using the approach presented by *De Groote et al. (2009)*, and we introduced additional control variables for the time derivatives of the activations $u_{\dot{a}}$. This again led to enforcing the first-order activation dynamics with simple constraints, and we enforced the activation dynamics equations by imposing two sets of linear inequality path constraints. These constraints can be derived from the original differential equation describing activation dynamics by using the upper and lower bounds of the excitations, which are often used as control variables. By formulating the activation dynamics in this manner we avoided the need to include excitations as control variables, and the resulting path constraints possessed favourable optimisation properties as they were linear.

For the purposes of modelling foot-ground interaction we attached four and two contact spheres to our model's right rearfoot and forefoot segments, respectively. The GRF generated by each of the contact spheres was based on a smooth foot-ground contact model (*Serrancolí et al., 2019*). The smooth foot-ground contact model was designed to be compatible with gradient-based optimal control approaches and is an approximation

of a previously published foot-ground contact model (*Sherman, Seth & Delp, 2011*). The equations used to calculate the normal and friction forces for the smooth foot-ground contact model can be accessed online at https://simbody.github.io/3.7.0/classSimTK_1_ 1SmoothSphereHalfSpaceForce.html. The position of each contact sphere relative to the segment it was attached to, and the stiffness and damping coefficients common to all the contact spheres were treated as parameters $p$ to be determined from the data-tracking calibration simulations. A constraint was added to ensure that the contact spheres lie on a horizontal plane when the ankle, subtalar and MTP DOFs were neutral. We used the default parameters for the constants responsible for smoothing the penetration and penetration rate terms, and the constant that enforces the derivative of the penetration term to be non-zero. For each contact sphere, the static, dynamic and viscous coefficients of friction were set to 0.95, 0.3 and 0.3, respectively, and the transition velocity and radius were set to 0.001 m/s and 0.02 m, respectively. We also introduced GRF control variables $u_{\mathrm{GRF}}$ for each of the contact spheres, similarly to *Serrancolí et al. (2019)*. GRF control variables were introduced to improve the conditioning of the objective function, as the GRF computed from the contact model is susceptible to large changes with only minor changes in the musculoskeletal model's state variables, which can lead to convergence issues. Equality path constraints were included to ensure the GRF from both the control variables and the contact model matched at the optimal solution.

The objective function $J$ consisted of three terms: a data-tracking term $J_{\mathrm{tracking}}$ that minimised the squared errors between the experimental and simulated kinematics, GRF and net joint moments, an effort term $J_{\mathrm{effort}}$ that minimised the squared activations, and a control variables term $J_{\mathrm{control}}$ that minimised the squared reserve actuators control variables and those control variables introduced due to the use of the implicit differential equations:

$$J = J_{\mathrm{tracking}} + J_{\mathrm{effort}} + J_{\mathrm{control}} \tag{1}$$

$$
\begin{aligned}
J_{\mathrm{tracking}} = {} & w_1 \sum_{j=1}^{37} \int_{t_0}^{t_f} \left( \frac{q_j^{\mathrm{EXP}} - q_j^{\mathrm{SIM}}}{\mathrm{range}\left(q_j^{\mathrm{EXP}}\right)} \right)^2 dt + w_1 \sum_{n=1}^{3} \int_{t_0}^{t_f} \left( \frac{\mathrm{GRF}_n^{\mathrm{EXP}} - u_{\mathrm{GRF_n}}^{\mathrm{SIM}}}{\mathrm{range}\left(\mathrm{GRF}_n^{\mathrm{EXP}}\right)} \right)^2 dt \\
& + w_2 \sum_{k=1}^{29} \int_{t_0}^{t_f} \left( \frac{\tau_k^{\mathrm{EXP}} - \tau_k^{\mathrm{SIM}}}{\mathrm{range}\left(\tau_k^{\mathrm{EXP}}\right)} \right)^2 dt
\end{aligned}
\tag{2}
$$

$$J_{\mathrm{effort}} = w_3 \sum_{i=1}^{92} \int_{t_0}^{t_f} \left( \frac{F_i^{\max} a_i^{\mathrm{SIM}^2}}{\sum_{i=1}^{92} F_i^{\max}} \right) dt \tag{3}$$

$$
\begin{aligned}
J_{\mathrm{control}} = {} & w_4 \sum_{m=1}^{17} \int_{t_0}^{t_f} \left( \frac{u_{\mathrm{res}_m}^{\mathrm{SIM}}}{\mathrm{bound}\left(u_{\mathrm{res}_m}^{\mathrm{SIM}}\right)} \right)^2 dt + w_5 \sum_{j=1}^{37} \int_{t_0}^{t_f} \left( \frac{u_{\dot{v}_j}^{\mathrm{SIM}}}{\mathrm{range}\left(\ddot{q}_j^{\mathrm{EXP}}\right)} \right)^2 dt \\
& + w_5 \sum_{i=1}^{92} \int_{t_0}^{t_f} \left( \frac{u_{\dot{F}_{T_i}}^{\mathrm{SIM}}}{\mathrm{bound}\left(u_{\dot{F}_{T_i}}^{\mathrm{SIM}}\right)} \right)^2 dt + w_5 \sum_{i=1}^{92} \int_{t_0}^{t_f} \left( \frac{u_{\dot{a}_i}^{\mathrm{SIM}}}{\mathrm{bound}\left(u_{\dot{a}_i}^{\mathrm{SIM}}\right)} \right)^2 dt
\end{aligned}
\tag{4}
$$

where the superscripts EXP and SIM denote experimental and simulated variables, respectively, $t_0$ and $t_f$ denote the initial and final times, respectively, of the experimental data from the trial being tracked, $\tau_k$ are the net joint moments, $F_i^{\max}$ are the MTU maximal isometric force parameters, and $w_i$ are the weighting factors whose values were set based upon the importance of the term being tracked or minimised ($w = [0.1\ 0.01\ 0.001\ 0.01\ 0.0001]$).

The objective function weighting factors were chosen heuristically, and we placed a greater emphasis on tracking the variables we believed were closer to the ground-truth (kinematics and GRF) whilst ensuring we obtained dynamically consistent solutions. To achieve those objectives it was necessary to place a lower weighting on the tracking of the net joint moments. Furthermore, we opted to not include the tracking of the net MTP moments, as we were not confident in the values obtained from IDA. We included the minimisation of the control variables introduced due to the use of the implicit differential equations with a very small weighting factor to avoid redundancy in the process of obtaining an optimal solution and to improve convergence. The experimental and simulated variable differences within $J_{\text{tracking}}$, except for the anterior-posterior pelvis translation difference, and the time derivatives of the generalised velocities control variables were each normalised by 10% of their respective experimental range from the trial being tracked. The anterior-posterior pelvis translation difference was normalised by 0.01 m to ensure it was tracked closely, since the range was much greater in comparison to the other kinematic variables. The remaining variables within $J_{\text{control}}$ were normalised by their respective upper bounds.

## Optimal control problem solution approach

We converted the data-tracking OCPs described in the previous section to discrete nonlinear programming problems (NLPs) using a direct collocation method. A *flipped* Legendre-Gauss-Radau direct collocation method (*Garg et al., 2011*) was used to discretise the time horizon of the data-tracking simulations across 80 equally spaced mesh intervals, and each mesh interval was further discretised with 4 points. We set the number of mesh intervals based on a previous study that performed tracking simulations of a running stride using a direct collocation method (*Lin & Pandy, 2017*), and by considering the trade-off between the time taken for an optimal solution to be reached and the influence of discretising the experimental data to be tracked for different numbers of mesh intervals. The state variables were parameterised with third-order Lagrange polynomials within each mesh interval. The control variables were parameterised at the beginning of each mesh interval and assumed to be piecewise constant throughout a mesh interval. The first-order dynamics constraints were enforced at the collocation points, whilst all the equality and inequality path constraints were enforced at the beginning of each mesh interval. We also included continuity constraints for the state variables between the ending and beginning of the mesh intervals.

Four data-tracking calibration simulations were performed in total. Three of these calibration simulations featured tracking experimental data from the first trial of each of the sprinting phases (early acceleration, mid-acceleration and maximum velocity)

independently whilst also determining the foot-ground contact model parameters (individual calibration simulations). In the fourth calibration simulation we tracked the experimental data from the same three trials simultaneously whilst also determining the foot-ground contact model parameters that were common across the trials (simultaneous calibration simulation). A data-tracking validation simulation was also performed, and we tracked the experimental data from the second maximum velocity phase trial whilst using the foot-ground contact model parameters obtained from the simultaneous calibration simulation. We selected to track the experimental data from the second maximum velocity phase trial for the validation simulation to demonstrate the worst-case scenario from a tracking perspective, as we noticed that the largest tracking errors for the calibration simulations were obtained from the first maximum velocity phase trial.

For each simulation, the initial guess alongside the lower and upper bounds of the state and control variables were set using the experimental data of the trial being tracked where possible. The generalised coordinates and velocities state variables, and the time derivatives of the generalised velocities control variables were initialised using the splined experimental kinematics. The initial guess for the upper-limb joint actuators control variables was set using the results from IDA. The reserve actuators and GRF control variables were initialised as zero for all the simulations. We used the same initial guess as *Falisse et al. (2019a)* for the activations and normalised tendon forces state variables, and the derivatives of the activations and normalised tendon forces control variables for all the simulations. The stiffness and damping parameters for the initial guess were set to 1e6 N/m$^2$ and 0.5 s/m, respectively, while the contact sphere position parameters were initialised so that they cover a plausible region of foot-ground contact during sprinting.

The lower and upper bounds of the state and control variables pertaining to the skeletal dynamics were set to be 25% less than and more than the minimum and maximum values, respectively, obtained from the experimental data. We set the lower and upper bounds of the variables pertaining to contraction and activation dynamics in accordance with those used by *Falisse et al. (2019a)*. The bounds of the anterior-posterior and medial-lateral positions of the contact sphere parameters were set as to prevent the centres of the spheres from overlapping and to lie within the boundaries of the rearfoot and forefoot segments. The upper bounds of the vertical positions of the contact sphere parameters were set to be the origin of the rearfoot and forefoot segments, whilst the lower bounds of those parameters were set by accounting for the length of the spikes attached to athlete's sprinting spikes, the sole height of the sprinting spikes and the behaviour of the compliant foot-ground contact model. The lower and upper bounds of the stiffness and damping parameters were set in line with those reported by previous studies using similar foot-ground contact models (*Falisse et al., 2019a*; *Lin & Pandy, 2017*; *Porsa, Lin & Pandy, 2016*; *Serrancolí et al., 2019*).

The data-tracking simulations were formulated in MATLAB (2016b; MathWorks Inc., Natick, MA, USA) using CasADi (*Andersson et al., 2019*), and solved using IPOPT (*Wächter & Biegler, 2006*) with an adaptive barrier parameter strategy and NLP convergence tolerance of $10^{-4}$. For each NLP, the variables were scaled to lie on the interval

[−1, 1] and we also scaled the constraints following the recommendations made by *Betts (2010)* to improve the convergence rate and numerical conditioning. As an example, the equality path constraints of the pelvis residual forces were scaled by the athlete's bodyweight (BW), thus the largest permissible violation for these constraints was 0.0071 N given our NLP convergence criteria. To further increase the computational efficiency of our simulations we used modified versions of OpenSim and Simbody (*Falisse et al., 2019a*) for the purposes of evaluating the skeletal dynamics equations. These versions are interfaced with CasADi, which permits the calculation of derivatives using algorithmic differentiation. To evaluate the data-tracking simulations we calculated the root mean squared difference (RMSD) between the tracked experimental and simulated data. For each of the RMSDs reported below we calculated the average RMSD for categories of variables (e.g. global pelvis angles and net lower-limb joint moments), and for the calibration simulations those categorical RMSDs were averaged across the three trials. The GRF RMSD components were expressed relative to the athlete's BW and as a percentage relative to the maximum absolute value of the GRF component obtained experimentally for the trial being tracked. As a further means of evaluating the data-tracking simulations, the filtered EMGs were compared to the corresponding simulated activations.

## RESULTS

The kinematics and GRF obtained from all the data-tracking simulations were a close match with the experimental kinematics (Figs. 3–5) and GRF (Figs. 6 and 7). For both sets of calibration simulations, the average RMSDs were less than 1° and 0.2 cm for the global pelvis angles and translations, respectively, and 1° for the relative joint angles (Table 1). Right ankle plantarflexion-dorsiflexion tended to exhibit the largest RMSD across each of the three trials and for both sets of calibration simulations (4.0° largest RMSD). For the validation simulation, the average RMSDs were less than 1° and 0.3 cm for the global pelvis angles and translations, respectively, and 1° for the relative joint angles. The largest kinematics tracking error for the validation simulation was also obtained for right ankle plantarflexion-dorsiflexion (5.5° RMSD).

For the GRF components, the average percentage RMSDs for both sets of calibration simulations (individual and simultaneous) were 8.2% and 8.0% (anterior-posterior), 4.2% and 4.1% (vertical) and 4.2% and 4.0% (medial-lateral). There was a noticeable trend for the tracking errors of the GRF components to increase from the early acceleration to the mid-acceleration trials and from the mid-acceleration to the maximum velocity trials for both sets of calibration simulations (Table 1). For the validation simulation, the percentage RMSDs of the GRF components were 11.4% (anterior-posterior), 5.9% (vertical) and 7.5% (medial-lateral).

The patterns of the net joint moments obtained from the individual and simultaneous data-tracking simulations were found to match the patterns of the experimental net joint moments (Figs. 8 and 9), with average RMSDs of 16.9 and 17.2 Nm, respectively. The average RMSDs of the net lower-limb joint moments were 19.5 and 19.9 Nm for the individual and simultaneous calibration simulations, respectively (Table 2). For the

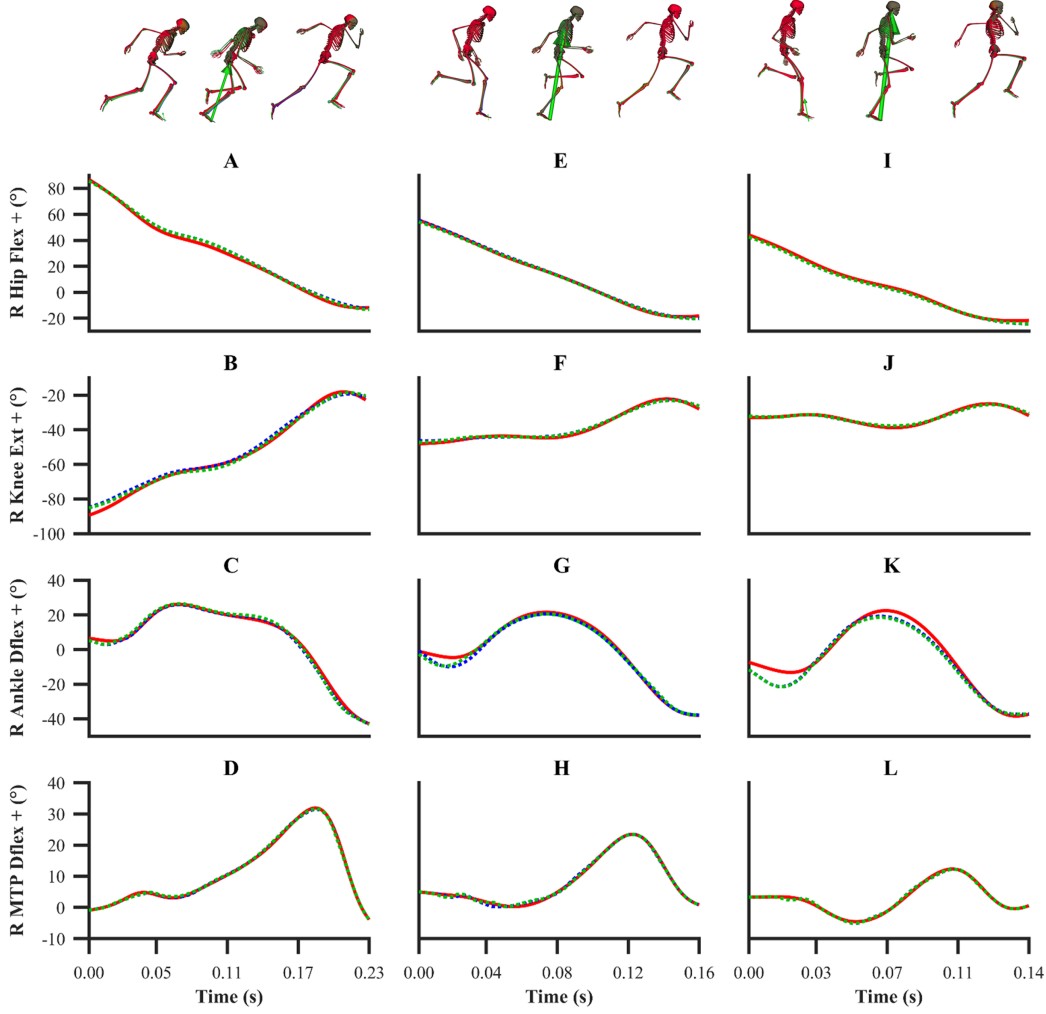

**Figure 3 Subset of right lower-limb joint angles for the first early acceleration (A–D), mid-acceleration (E–H) and maximum velocity (I–L) phase trials.** Experimental joint angles are denoted by solid red lines. Simulated joint angles from the individual and simultaneous calibration simulations are denoted by dashed blue and green lines, respectively.

validation simulation, the average RMSDs of the net joint moments and the net lower-limb joint moments were 17.7 and 23.6 Nm, respectively. The tracking errors for the validation simulation were lower than the corresponding tracking errors obtained from the individual (24.9 and 28.6 Nm RMSD) and simultaneous (24.9 and 28.7 Nm RMSD) calibration simulations of the maximum velocity phase trial. For the calibration simulations, the pattern of the untracked net right MTP plantarflexion-dorsiflexion moment was markedly different around touchdown and mid-stance (18.2 Nm largest RMSD), whilst for the validation simulation it was in closer agreement (8.7 Nm RMSD).

The simulated activations of the GASTM, TFL, VM and SOL for both sets of calibration simulations displayed similarities in terms of magnitude and timing with the corresponding EMG data (Fig. 10). The simulated GMAX and BF activations from both sets of calibration simulations were markedly different with respect to the

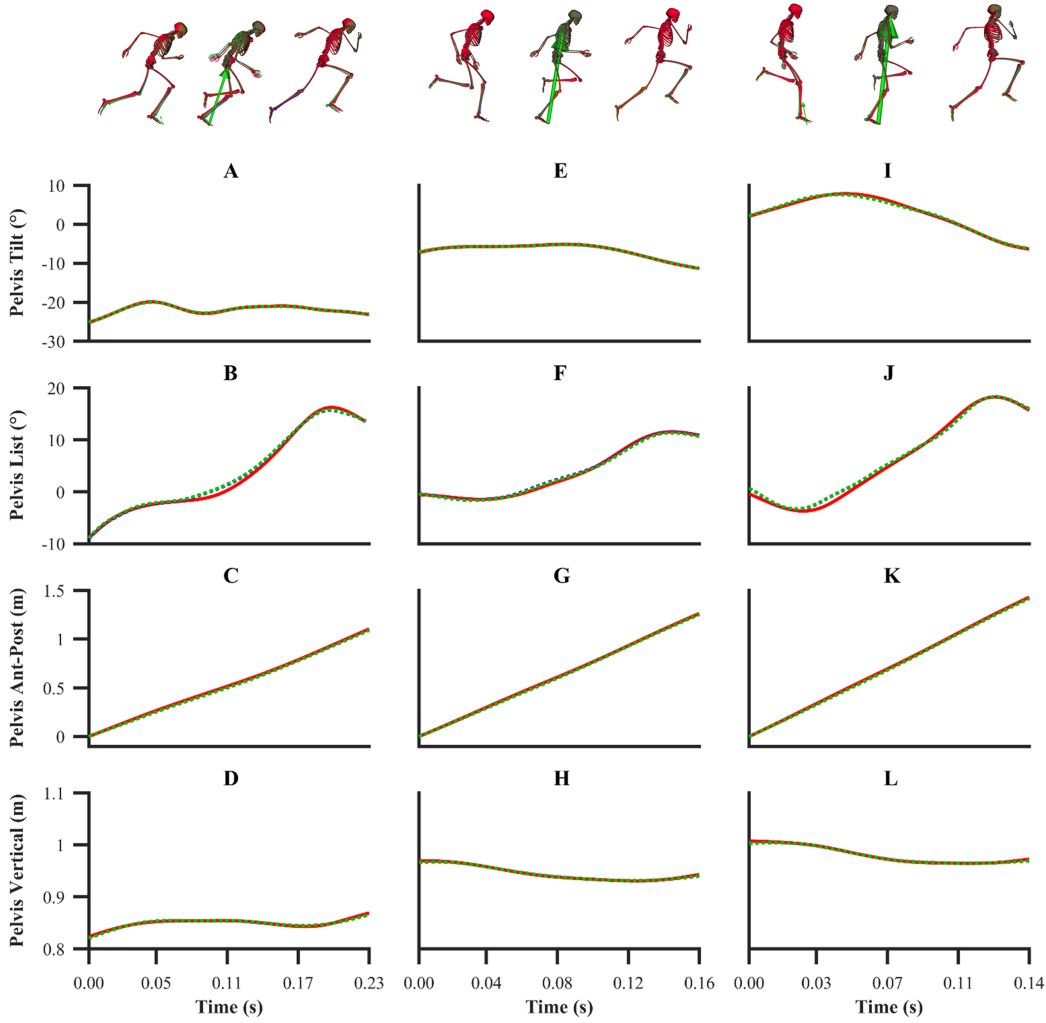

**Figure 4 Subset of global pelvis angles and translations for the first early acceleration (A–D), mid-acceleration (E–H) and maximum velocity (I–L) phase trials.** Experimental global pelvis angles and translations are denoted by solid red lines. Simulated global pelvis angles and translations from the individual and simultaneous calibration simulations are denoted by dashed blue and green lines, respectively.

corresponding EMG data. A noticeable difference during the end of the stance phase and the beginning of the following flight phase was observed between the simulated activations and EMG data of the GMAX, BF and GASTM, in which the simulated activations continued to ramp up or remain constant, whilst the EMG data tended to zero. For the validation simulation, the simulated activations of the BF, GASTM and VM displayed strong similarities with the corresponding EMG data (Fig. 11). The simulated BF and GASTM activations for the validation simulation, as per the calibration simulations, continued to ramp up near the end of the stance phase and the beginning of the following flight phase, whilst the corresponding EMG data tended to zero. The simulated GMAX activation for the validation simulation displayed similarities to the corresponding EMG data, although the magnitude was discernibly different.
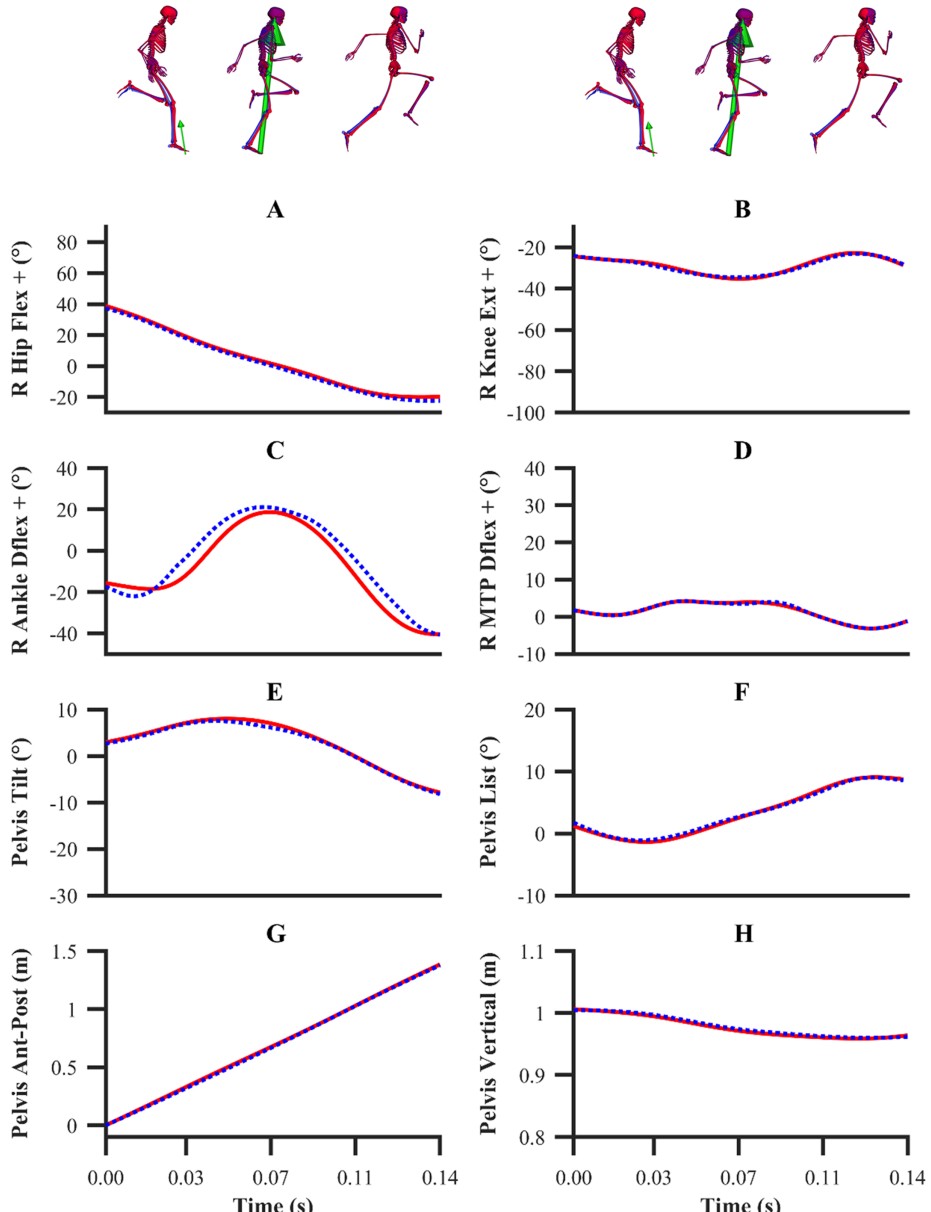

**Figure 5 Subset of right lower-limb joint angles (A–D), and global pelvis angles and translations (E–H) for the second maximum velocity phase trial.** Experimental joint angles, and global pelvis angles and translations are denoted by solid red lines. Simulated joint angles, and global pelvis angles and translations from the validation simulation are denoted by dashed blue lines.

The optimised sphere position parameters were for the most part reasonably similar when obtained from the individual and simultaneous calibration simulations (Table 3; Fig. 12). The optimised stiffness parameters displayed a trend of increasing from the early acceleration phase trial to the maximum velocity phase trial when determined from the individual calibration simulations. For the simultaneous calibration simulation, the optimised stiffness parameter was similar to the average value determined from the

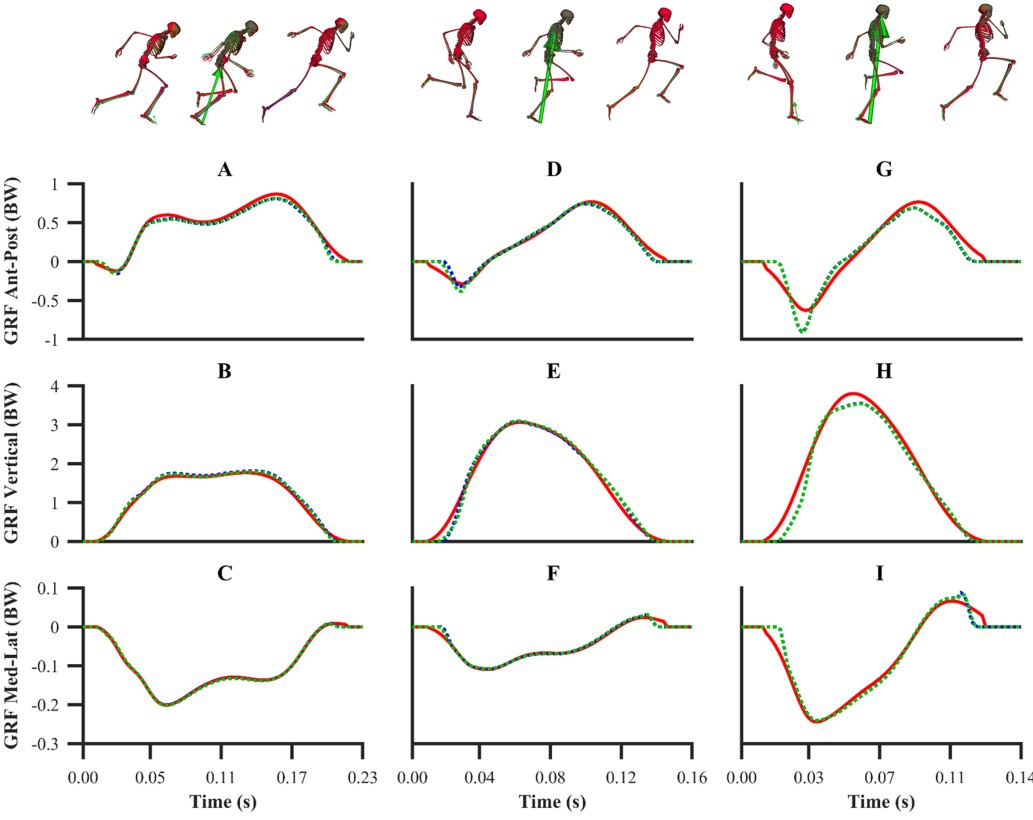

**Figure 6 Normalised GRF components for the first early acceleration (A–C), mid-acceleration (D–F) and maximum velocity (G–I) phase trials.** Experimental GRF components are denoted by solid red lines. Simulated GRF components from the individual and simultaneous calibration simulations are denoted by dashed blue and green lines, respectively.

individual calibration simulations. The optimised damping parameters were found to change minimally between the individual and simultaneous calibration simulations.

## DISCUSSION

The primary aims of the current study were to develop a musculoskeletal modelling and simulation framework for sprinting, and to evaluate its capability of reproducing highly dynamic experimental data. The secondary aims of this study were to generate dynamically consistent simulated outputs and to identify foot-ground contact model parameters for subsequent predictive simulations. To achieve our aims, we performed a series of data-tracking calibration and validation simulations, based upon a direct collocation optimal control approach. The data-tracking calibration simulations also enabled the determination of the foot-ground contact model parameters from tracking either an individual trial or multiple trials simultaneously. We found that the outputs from the calibration and validation simulations closely matched the experimental data, and this provides confidence in using the framework to address applied sprinting research questions. Importantly, all the simulated outputs were dynamically consistent, implying that no fictious residual forces and moments were necessary to satisfy the dynamics equations of our musculoskeletal model. The proposed framework also includes a novel

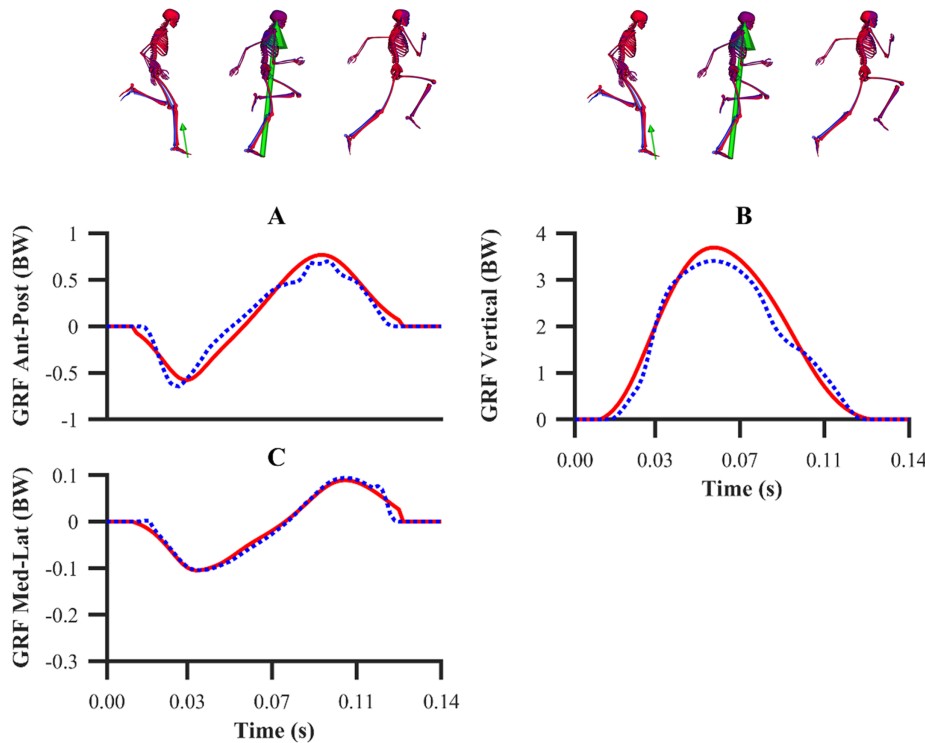

**Figure 7 Normalised GRF components for the second maximum velocity phase trial (A–C).** Experimental GRF components are denoted by solid red lines. Simulated GRF components from the validation simulation are denoted by dashed blue lines.

contribution to the biomechanics modelling and simulation literature, as it enables dynamically consistent simulations of an explosive locomotor task to be performed (with the aerodynamic drag force included), whilst also identifying foot-ground contact model parameters. Furthermore, the simultaneous data-tracking calibration simulation enabled the foot-ground contact model parameters to be determined from multiple trials, which is key to avoid overfit bias.

The smooth foot-ground contact model used within our framework has been previously shown to be appropriate for reproducing the GRF within walking (*Falisse et al., 2019a*) and a sit-to-stand task (*Serrancolí et al., 2019*). In this study, we showed for the first time that the smooth foot-ground contact model permits accurate reproduction of the GRF from different sprinting phases. Average RMSDs of less than 0.07, 0.15 and 0.01 BW for the anterior-posterior, vertical and medial-lateral GRF components, respectively, across all the simulations performed are comparable with another study (*Lin & Pandy, 2017*), in which similar methods were used to perform dynamically consistent data-tracking simulations of running. In our study we also observed that the GRF tracking errors were the highest for the maximum velocity phase trials, which is likely due to the rapid change in GRF dynamics in comparison to the early acceleration phase trial. This result has also been observed by *Lin & Pandy (2017)*, who reported lower GRF tracking error in walking compared to running trials. Additionally, the simulated GRFs we obtained were noticeably smoother than those obtained by *Lin & Pandy (2017)*, which

**Table 1 The root mean squared difference (RMSD) of the global pelvis translations and angles, subset of lower-limb joint angles and normalised GRF components.**

| RMSD | Early acc | | Mid-acc | | Max vel | | Max vel |
|---|---|---|---|---|---|---|---|
| | Indiv | Simult | Indiv | Simult | Indiv | Simult | Valid |
| *Pelvis Translations* | | | | | | | |
| Pelvis Ant-Post (cm) | 0.4 | 0.4 | 0.3 | 0.3 | 0.4 | 0.4 | 0.4 |
| Pelvis Vertical (cm) | 0.1 | 0.2 | 0.1 | 0.1 | 0.1 | 0.1 | 0.2 |
| Pelvis Med-Lat (cm) | 0.1 | 0.1 | 0.1 | 0.1 | 0.1 | 0.1 | 0.1 |
| *Pelvis Angles* | | | | | | | |
| Pelvis List (°) | 0.1 | 0.1 | 0.1 | 0.1 | 0.3 | 0.3 | 0.4 |
| Pelvis Tilt (°) | 0.5 | 0.6 | 0.2 | 0.2 | 0.3 | 0.3 | 0.3 |
| Pelvis Rotation (°) | 1.0 | 1.1 | 0.7 | 0.7 | 0.8 | 0.8 | 0.6 |
| *Lower-limb Joint Angles* | | | | | | | |
| R Hip Flex (°) | 1.8 | 1.7 | 0.6 | 0.7 | 1.3 | 1.3 | 1.4 |
| R Knee Ext (°) | 2.0 | 1.6 | 0.9 | 0.8 | 0.6 | 0.6 | 0.5 |
| R Ankle Dflex (°) | 1.2 | 1.5 | 2.0 | 1.9 | 3.7 | 4.0 | 5.5 |
| R MTP Dflex (°) | 0.3 | 0.3 | 0.4 | 0.4 | 0.3 | 0.3 | 0.2 |
| L Hip Flex (°) | 2.7 | 2.7 | 1.5 | 1.3 | 1.4 | 1.5 | 2.7 |
| L Knee Ext (°) | 0.3 | 0.4 | 0.2 | 0.2 | 0.6 | 0.6 | 1.4 |
| L Ankle Dflex (°) | 0.3 | 0.3 | 0.2 | 0.2 | 0.2 | 0.2 | 0.2 |
| L MTP Dflex (°) | 1e−2 | 1e−2 | 1e−2 | 1e−2 | 1e−2 | 1e−2 | 1e−2 |
| *GRF* | | | | | | | |
| Ant-Post (BW) | 0.042 | 0.045 | 0.047 | 0.041 | 0.103 | 0.102 | 0.088 |
| Vertical (BW) | 0.058 | 0.049 | 0.112 | 0.122 | 0.213 | 0.213 | 0.219 |
| Med-Lat (BW) | 0.002 | 0.003 | 0.005 | 0.005 | 0.016 | 0.016 | 0.008 |

**Note:**
The values presented are from the individual (Indiv) and simultaneous (Simult) calibration simulations of the first early acceleration (Early Acc), mid-acceleration (Mid-Acc) and maximum velocity (Max Vel) phase trials, and from the validation (Valid) simulation of the second maximum velocity phase trial.

exhibited unrealistic transients. This can potentially be attributed to the generic foot-ground contact model parameters used in that study, whilst in the current study those parameters were determined from the data-tracking calibration simulations. This study therefore adds to the existing literature which has demonstrated that the determination of the foot-ground contact model parameters via data-tracking simulations enables realistic GRF to be obtained (*Falisse et al., 2019a*; *Serrancolí et al., 2019*).

The ability to track several different trials simultaneously permits a model's parameters to be obtained with the confidence that they are less likely to suffer from overfitting. We observed that the individual data-tracking calibration simulations resulted in only minor improvements in terms of GRF tracking error when compared to those obtained from the simultaneous data-tracking calibration simulation, thus demonstrating the foot-ground contact model's ability to generalise across different sprinting phases. As a means of further evaluating our framework, we performed a data-tracking validation simulation using the foot-ground contact model parameters obtained from the simultaneous data-tracking calibration simulation. The low GRF tracking errors shown in

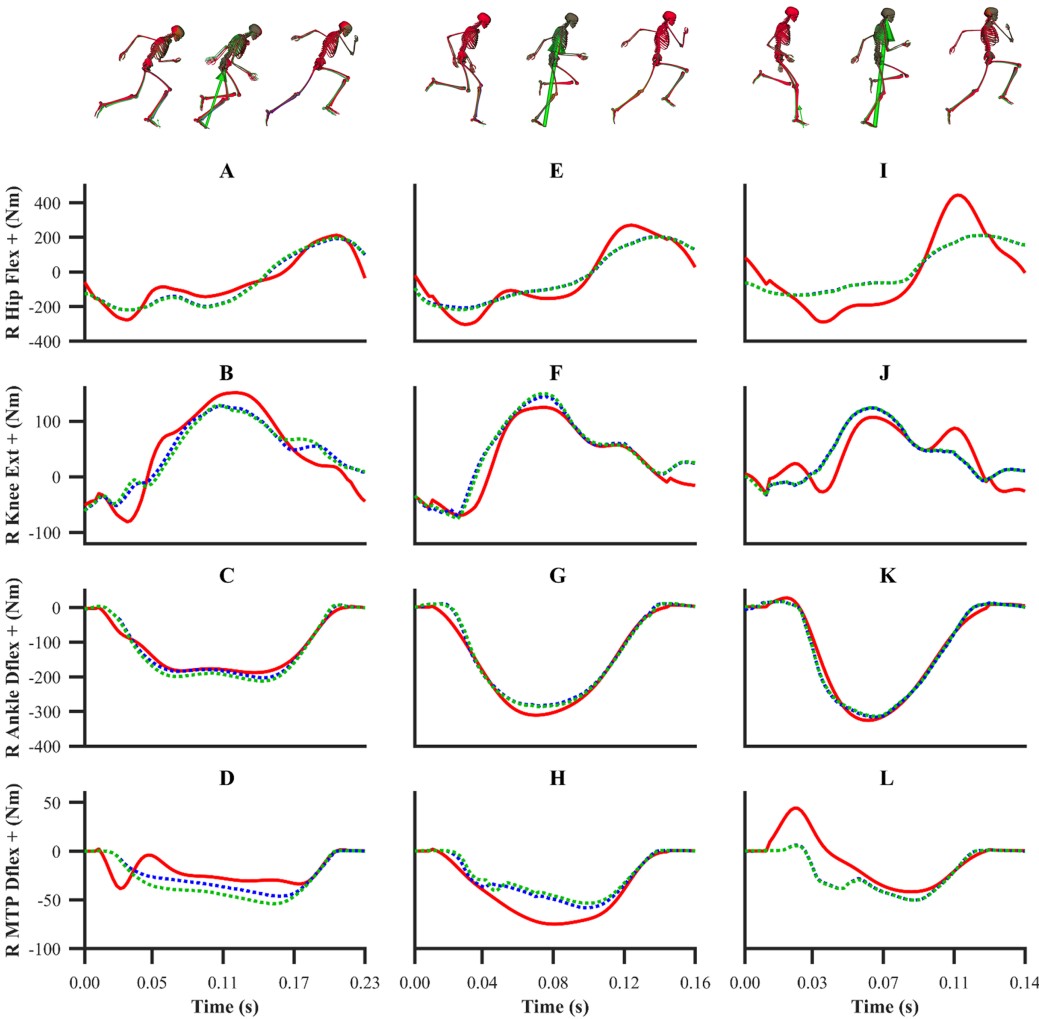

**Figure 8 Subset of right lower-limb net joint moments for the first early acceleration (A–D), mid-acceleration (E–H) and maximum velocity (I–L) phase trials.** Net MTP moments were not tracked. Experimental net joint moments are denoted by solid red lines. Simulated net joint moments from the individual and simultaneous calibration simulations are denoted by dashed blue and green lines, respectively.                                                               

the simultaneous calibration and validation simulations provide confidence in using the foot-ground contact model parameters obtained from the simultaneous data-tracking calibration simulation to perform future predictive simulations of sprinting. It is also worth highlighting that the formulation of a simultaneous data-tracking calibration simulation is greatly facilitated by the direct collocation method and requires minimal adjustments to the NLP problem formulation of tracking a single trial (increase in the number of variables and constraints). Future studies should therefore consider using a similar approach for circumstances in which the parameters cannot be easily obtained empirically.

From a kinematics perspective, the magnitude of the tracking errors across all simulations are in line with the study by *Lin & Pandy (2017)*, who reported RMSDs of less than 1.5° and 0.6 cm for rotational and translational tracked kinematics during
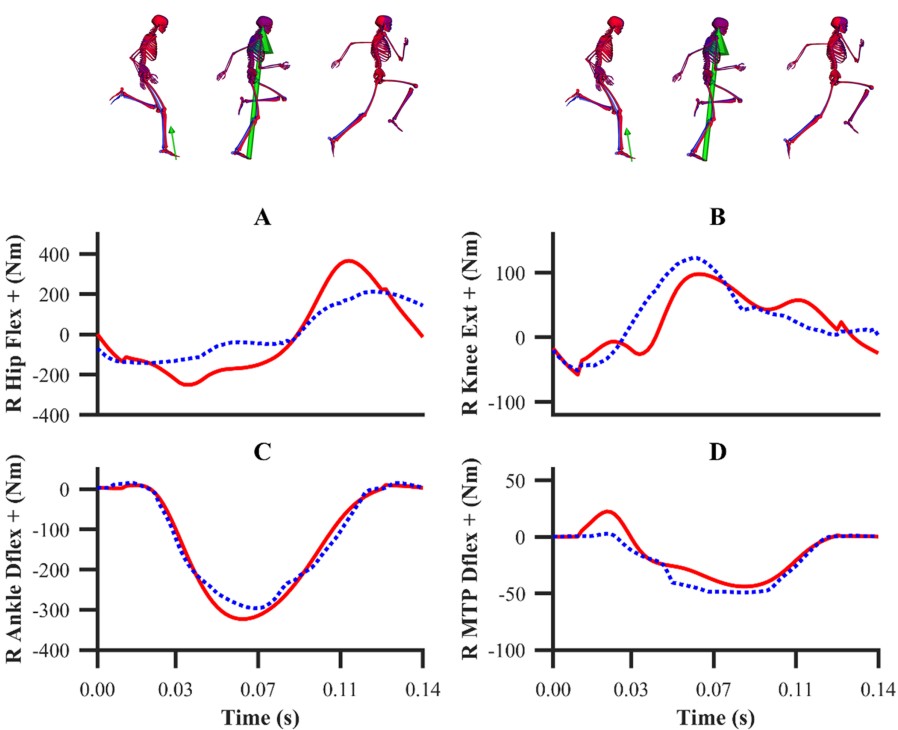

**Figure 9 Subset of right lower-limb net joint moments for the second maximum velocity phase trial (A–D).** Net MTP moments were not tracked. Experimental net joint moments are denoted by solid red lines. Simulated net joint moments from the validation simulation are denoted by dashed blue lines.

**Table 2 The root mean squared difference (RMSD) of a subset of lower-limb net joint moments.**

|  | Early acc | | Mid-acc | | Max vel | | Max vel |
| --- | --- | --- | --- | --- | --- | --- | --- |
| RMSD | Indiv | Simult | Indiv | Simult | Indiv | Simult | Valid |
| Lower-limb net joint moments (Nm) | | | | | | | |
| R Hip Flex | 46.9 | 49.8 | 58.2 | 57.0 | 124.2 | 123.5 | 99.4 |
| R Knee Ext | 26.1 | 30.3 | 21.8 | 21.4 | 29.1 | 28.5 | 30.2 |
| R Ankle Dflex | 12.3 | 18.0 | 16.9 | 16.3 | 14.3 | 14.1 | 18.8 |
| R MTP Dflex | 11.9 | 16.9 | 13.6 | 15.9 | 18.0 | 18.2 | 8.7 |
| L Hip Flex | 33.7 | 34.1 | 25.9 | 26.5 | 42.4 | 44.2 | 40.5 |
| L Knee Ext | 6.4 | 7.1 | 5.0 | 4.4 | 12.7 | 12.3 | 16.1 |
| L Ankle Dflex | 0.5 | 0.4 | 0.3 | 0.3 | 0.8 | 0.9 | 0.7 |
| L MTP Dflex | 2e−2 | 2e−2 | 2e−2 | 2e−2 | 5e−2 | 6e−2 | 5e−2 |

Note:
Net MTP moments were not tracked. The values presented are from the individual (Indiv) and simultaneous (Simult) calibration simulations of the first early acceleration (Early Acc), mid-acceleration (Mid-Acc) and maximum velocity (Max Vel) phase trials, and from the validation (Valid) simulation of the second maximum velocity phase trial.

running, respectively. In general, the patterns of simulated kinematics we obtained are in agreement with previous experimental sprinting kinematics (*Lai, 2015*; *Von Lieres und Wilkau, 2017*), and the kinematics errors we observe are in the same order of magnitude of the experimental error related to human motion capture measurement (*Alenezi et al.,*

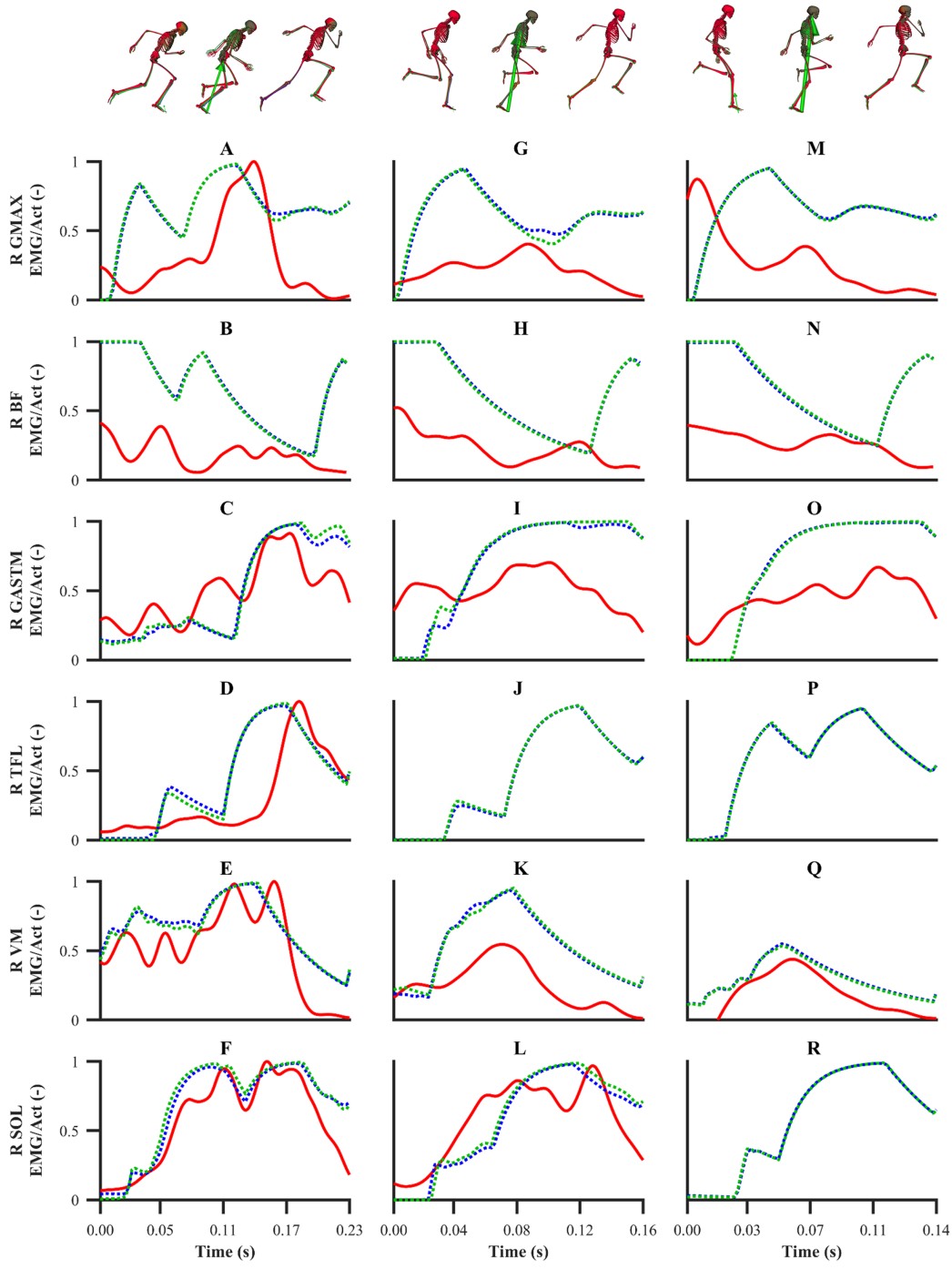

**Figure 10 EMGs and simulated activations from the right lower-limb for the first early acceleration (A–F), mid-acceleration (G–L) and maximum velocity (M–R) phase trials.** The TFL EMG during the mid-acceleration and maximum velocity phase trials, and the SOL EMG during the maximum velocity phase trial were omitted due to data collection issues. EMGs are denoted by solid red lines. Simulated activations from the individual and simultaneous calibration simulations are denoted by dashed blue and green lines, respectively. GMAX, gluteus maximus; BF, biceps femoris; GASTM, gastrocnemius; TFL, tensor fasciae latae; VM, vastus medialis; SOL, soleus.

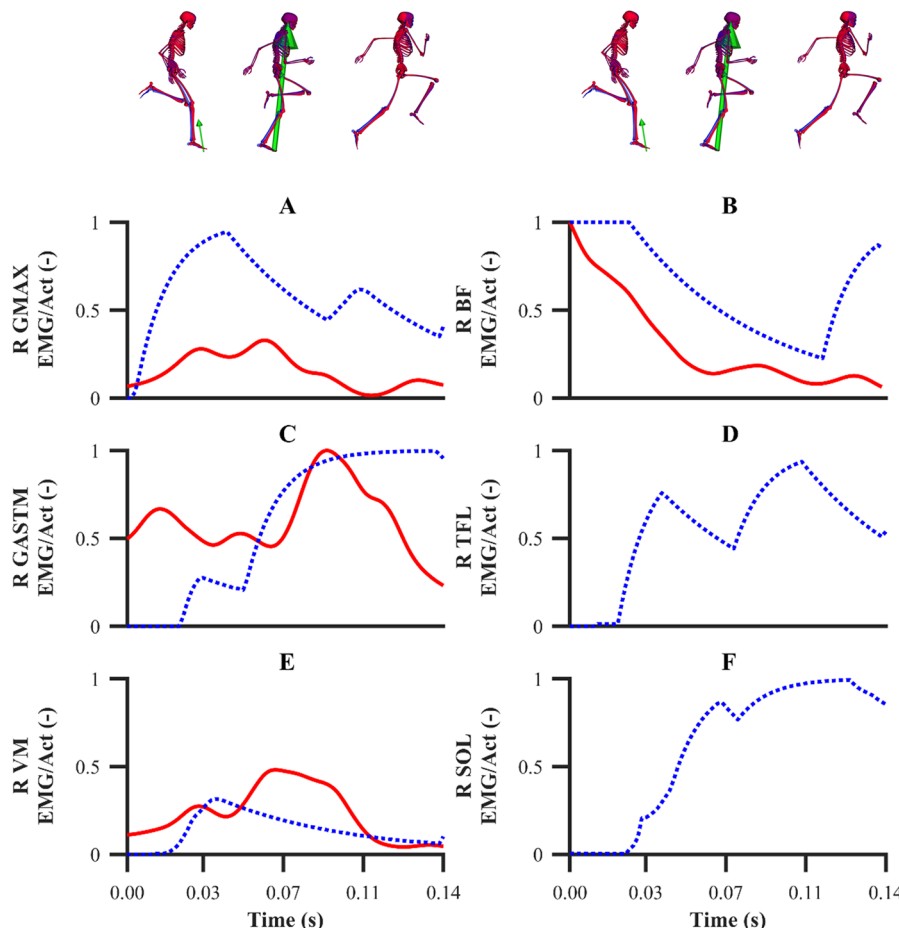

**Figure 11 EMGs and simulated activations from the right lower-limb for the second maximum velocity phase trial (A–F).** The TFL and SOL EMGs were omitted due to data collection issues. EMGs are denoted by solid red lines. Simulated activations from the validation simulation are denoted by dashed lines. GMAX, gluteus maximus; BF, biceps femoris; GASTM, gastrocnemius; TFL, tensor fasciae latae; VM, vastus medialis; SOL, soleus.

*2016*). We therefore believe that the current framework is of a sufficient accuracy to warrant its use within applied sprinting settings. An aspect of our framework which possibly warrants further improvement is the tracking of right ankle plantarflexion-dorsiflexion, for which the maximum RMSD was 5.5° and it was obtained from the data-tracking validation simulation. We were surprised by this result, as we anticipated the kinematics tracking errors from the validation simulation to be more evenly distributed amongst all the DOFs and higher than those obtained from the equivalent sprinting phase trial of the calibration simulations. In any case, it is worthwhile noting that the smallest detectable difference for peak ankle dorsiflexion during the stance phase of running and bend sprinting is in excess of 10° (*Alenezi et al., 2016*; *Judson et al., 2020*), and the largest right ankle dorsiflexion difference we obtained during the stance phase of the validation simulation was 8.9°. Interpreting these findings together suggests that the right ankle kinematics tracking error obtained is likely within the bounds of biological variation and measurement error.

**Table 3 Optimised foot-ground contact model parameters determined from the individual and simultaneous data-tracking calibration simulations.**

| Parameter | Early acceleration | Mid-acceleration | Maximum velocity | Simultaneous |
|---|---|---|---|---|
| Sphere $s_{1x}$ (m) | 0.070 | 0.066 | 0.069 | 0.070 |
| Sphere $s_{1y}$ (m) | −0.030 | −0.030 | −0.030 | −0.030 |
| Sphere $s_{1z}$ (m) | −0.028 | −0.020 | −0.030 | −0.029 |
| Sphere $s_{2x}$ (m) | 0.010 | 0.035 | 0.031 | 0.029 |
| Sphere $s_{2y}$(m) | −0.030 | −0.030 | −0.030 | −0.030 |
| Sphere $s_{2z}$ (m) | 0.045 | 3.07e−7 | 3.04e−7 | 1.52e−8 |
| Sphere $s_{3x}$ (m) | 0.150 | 0.150 | 0.150 | 0.150 |
| Sphere $s_{3y}$ (m) | −0.028 | −0.028 | −0.028 | −0.028 |
| Sphere $s_{3z}$ (m) | 6.46e−6 | 5.41e−5 | 1.12e−6 | 6.88e−8 |
| Sphere $s_{4x}$ (m) | 0.150 | 0.150 | 0.150 | 0.150 |
| Sphere $s_{4y}$ (m) | −0.028 | −0.028 | −0.028 | −0.028 |
| Sphere $s_{4z}$ (m) | −0.025 | −4.79e−5 | −0.022 | −0.023 |
| Sphere $s_{5x}$ (m) | 0.120 | 0.070 | 0.070 | 0.070 |
| Sphere $s_{5y}$ (m) | −0.028 | −0.028 | −0.028 | −0.028 |
| Sphere $s_{5z}$ (m) | 9.70e−6 | 0.045 | 0.045 | 0.045 |
| Sphere $s_{6x}$ (m) | 0.120 | 0.120 | 0.120 | 0.120 |
| Sphere $s_{6y}$ (m) | −0.028 | −0.028 | −0.028 | −0.028 |
| Sphere $s_{6z}$ (m) | −0.025 | −2.08e−6 | −9.32e−6 | −6.09e−8 |
| Stiffness (N/m$^2$) | 1.17e6 | 1.64e6 | 1.79e6 | 1.60e6 |
| Damping (s/m) | 0.073 | 0.110 | 0.063 | 0.072 |

Note:
Contact spheres $s_1$ and $s_2$ were attached to the forefoot segment, the rest were attached to the rearfoot segment. The $x$, $y$ and $z$ subscripts for the contact spheres correspond to the anterior-posterior, vertical and medial-lateral axes of the segments to which the spheres were attached.

A wide range of values for the lower-limb net joint moments have been reported within the sprinting literature. For example, the peak net hip flexion-extension moment has been reported to range between ~180 and ~750 Nm during maximum velocity sprinting (*Bezodis, Kerwin & Salo, 2008*; *Schache et al., 2019*), whilst in the current study the peak value was ~400 Nm. While it is possible that these differences may be attributed to an individual athlete's characteristics (e.g. individual strength differences and anthropometrics) or differences in performance levels, they are also likely to be influenced by the experimental data processing methods and modelling assumptions applied to determine the net joint moments. We therefore opted to track the net joint moments with a lower weighting compared to the kinematics and GRF, which were considered as the most reliable data to track, due to the various uncertainties known to influence the calculation of net joint moments from an IDA (*Derrick et al., 2020*). Despite the choice to lower the weight of the net joint moments tracking term, we obtained simulated net joint moments that largely followed the general patterns of the net joint moments we determined from IDA and of those reported within the existing sprinting IDA literature (*Bezodis, Kerwin & Salo, 2008*; *Schache et al., 2019*). We would also like to emphasise that the simulated net joint moments we obtained were from dynamically consistent

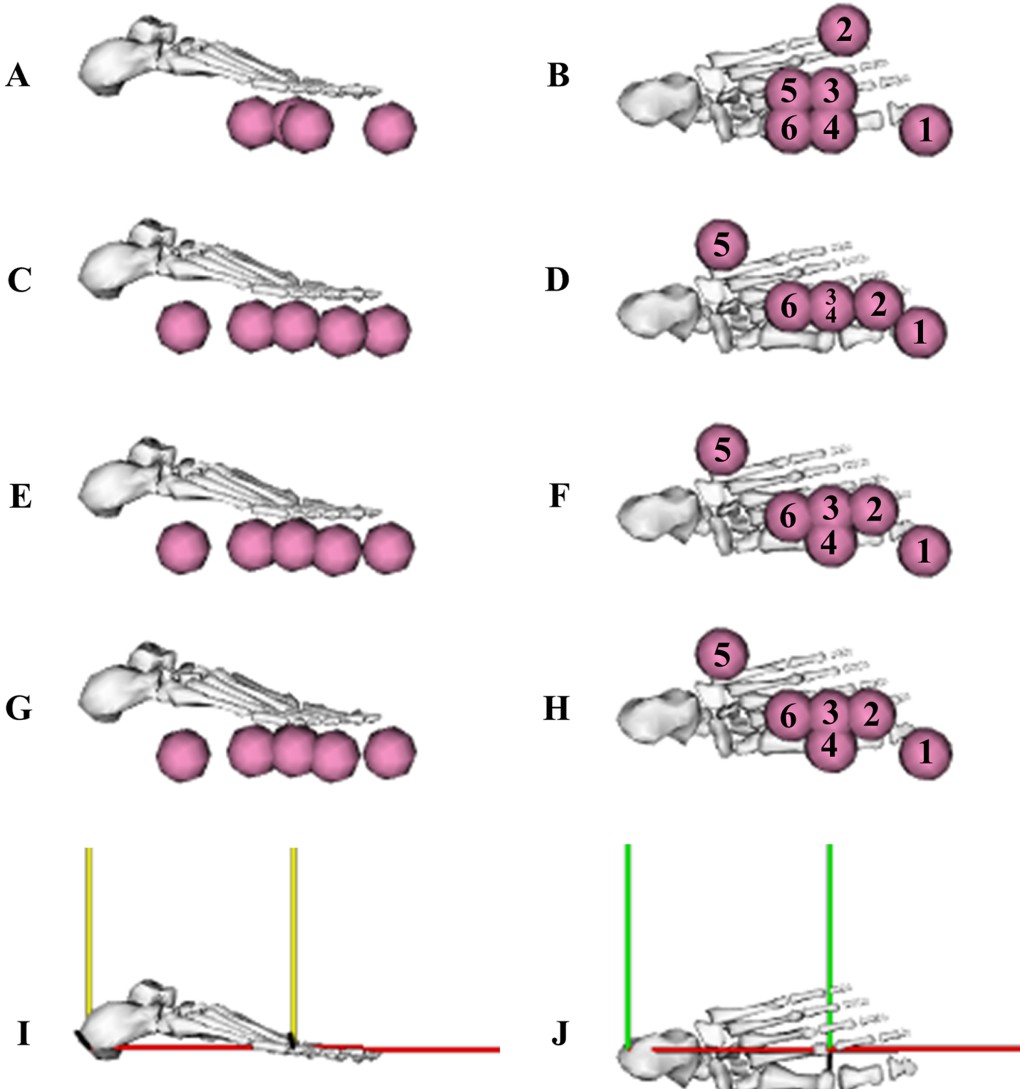

**Figure 12 Optimised contact sphere locations determined from the individual and simultaneous data-tracking calibration simulations.** Sagittal and transverse plane views of the optimised contact sphere locations from the first individual early acceleration phase trial (A and B), mid-acceleration phase trial (C and D), maximum velocity phase trial (E and F) calibration simulations, and for the three simultaneous trials calibration simulation (G and H). Contact sphere numbers, and rearfoot and forefoot segment axes (I and J) have been included to aid with the interpretation of the optimised locations presented in Table 3 (red axis: anterior-posterior ($x$), yellow axis: vertical ($y$) and green axis: medial-lateral ($z$)).

simulations, and they were potentially more physiologically plausible due to the use of Hill-type MTUs.

In our data-tracking simulations we avoided tracking the net MTP moments, as we felt that the right net MTP moment was unreliable due to the difficulty to distribute the GRF between the rearfoot and forefoot segments. We believed that the right net MTP moments obtained from all the data-tracking simulations were more physiologically plausible and realistic in comparison to those obtained from IDA, which exhibited
fluctuations during the first early acceleration and maximum velocity phase trials. Despite the challenges with calculating the net MTP moment from IDA, the MTP has been previously shown to undergo a substantial range of motion during the contact phase of sprinting (*Smith, Lake & Lees, 2014*). We therefore included a forefoot segment within our model to permit MTP motion. To date, most modelling and simulation studies have chosen to exclude modelling MTP motion, particularly those in which the model's lower-limbs are actuated by Hill-type MTUs (*Dorn, Schache & Pandy, 2012*; *Lai et al., 2016*). It is not necessarily surprising that previous studies have neglected modelling MTP motion during sprinting, as existing off the shelf musculoskeletal models, similar to the one we used, do not have the capacity to produce MTP moments that are in a similar magnitude to those obtained from isometric dynamometry testing (*Goldmann et al., 2013*). In our study, we set the upper limit of the MTP reserve actuators to 40 Nm, such that together with the Hill-type MTUs contained within the original model, they were able to better reflect the moment capacity at the MTP DOFs in relation to the existing literature. Future studies should consider exploring how to better model the MTP moment generating components to minimise the reserve actuator reliance.

An encouraging aspect our study was the similarities between the EMGs and the corresponding simulated activations. We were surprised by the similarities obtained considering the cost function used to resolve the MTU force-sharing problem, minimisation of weighted squared activations, not capturing the time-dependent performance criteria of sprinting. Discrepancies were observed during the end of the stance phase and the beginning of the following flight phase between the simulated activations and EMGs of the BF and GASTM across all the simulations performed. The calibration of the MTU parameters by performing data-tracking simulations and/or the tracking of the EMGs may lead to removing the anomalous simulated activations not observed in the EMGs. The direct collocation optimal control approach we have used in this study also enables the MTU force-sharing problem to be resolved using a cost function that more closely captures the time-dependent performance criteria of sprinting, such as MTU power (*Cavagna, Komarek & Mazzoleni, 1971*), and this may also reduce the aforementioned discrepancies.

The major benefit of the modelling and simulation approach used in this study is that it permits dynamically consistent simulations to be obtained. It is not uncommon to obtain large dynamic inconsistencies in the form of pelvis residual forces and moments that surpass 1,000 N and 300 Nm, respectively, when performing a standard IDA within sprinting (*Aeles et al., 2018*). The approach we have used in this study builds on work of other studies (*Lin & Pandy, 2017*; *Pallarès-López et al., 2019*) who demonstrated the ability of the direct collocation optimal control approach to reduce residuals within sporting tasks (hopping and running). In this study we demonstrated that it is possible to use this approach to generate dynamically consistent motions in a demanding task that spanned all three planes of motion. Consequently, we recommend that researchers within the sports biomechanics community adopt the approach we have used to increase the

fidelity of their results and obtain dynamically consistent motions. A limitation of our study is that we used a generic linearly scaled model. We suspect that the use of a subject-specific model would lead to reduced residuals when performing a standard IDA, due to the model having more representative inertial properties and mass distribution, and we expect that this would improve the tracking of the experimental data due to not needing to overcome the sizeable residuals obtained when using a linearly scaled generic model.

An alternative approach to generating motions that are closer to being dynamically consistent is RRA, which is included within OpenSim (*Delp et al., 2007*). RRA is based on using a multibody model actuated by a combination of joint and residual actuators, with the objective to track experimental kinematics whilst also minimising the use of the actuators. The reserve actuators are weighted heavily in comparison to the joint actuators such that the new motion obtained is closer to being dynamically consistent. A beneficial feature of RRA is that it provides recommendations for minimally adjusting the mass properties of the model, and this is a useful feature in circumstances when there are discrepancies between the model's and subject's mass distribution. RRA can then be performed again with the model in which the adjustments to the mass properties have been applied to further refine the dynamic consistency. A similar feature could also be incorporated within the data-tracking optimal control approach used in this study, similarly to how the foot-ground contact model parameters were determined. The downsides of RRA are that it must be performed iteratively to obtain a worthwhile dynamically consistent motion and that it necessitates explicit forward integration to satisfy the dynamics, which can be affected by integration errors. In contrast, the direct collocation optimal control approach used in this study needs only to be performed once to obtain a dynamically consistent solution, and it imposes the dynamics for each time step simultaneously as the state variables are treated as design variables alongside the control variables.

The most challenging elements to simulate were the transitions between touchdown and take-off. For all the data-tracking simulations performed, we observed discrepancies between the simulated and experimental right lower-limb kinematics and GRF around touchdown and take-off, and these discrepancies were most pronounced for the tracked maximum velocity phase trials (Figs. 6 and 7). Noticeable differences in either knee flexion-extension or ankle plantarflexion-dorsiflexion were accompanied by a mismatch in GRF production timings (largest touchdown and take-off mismatches were 8.0 and 12.6 ms, respectively). More specifically, the right lower-limb appeared to extend earlier and further in the simulations in comparison to the experimental kinematics. This can be explained by the model attempting to position the foot closer to the ground to allow the foot-ground contact model, which is driven by the kinematics, to generate the matching GRF. Future work exploring the geometry of the foot-ground contact model, similar to *Ezati et al. (2020)*, is possibly needed to improve the timing of GRF production.

Another explanation for the GRF timing differences may concern the filtering we used to process the experimental GRF. We opted to filter both the experimental kinematics

and GRF with the same cut-off frequency as per the current recommendations (*Derrick et al., 2020*). This was to avoid tracking oscillatory net joint moments, which cannot be produced by muscles as they are unable to activate/deactivate instantaneously. However, in this approach the cut-off frequency to filter the GRF is too low based on a residual analysis, which leads to artificially extending the ground contact phase (*Mai & Willwacher, 2019*; *Robertson & Dowling, 2003*). Thus, the GRF onset/offset timing differences, in addition to the right lower-limb extension, can potentially be explained by the tracking of the filtered GRF. In fact, the kinematics during the contact phases are tracked closely, particularly for the calibration simulations, which supports this explanation. A further possibility to improve the GRF timings involves filtering the experimental data using matching time varying cut-off frequencies established from the kinematics data (*Davis & Challis, 2020*), in an attempt to limit the onset/offset discrepancies between the filtered and raw experimental GRF data. It is important to note that filtering the kinematics and GRF data with matching time-varying frequencies may also lead to oscillatory net joint moments when performing a standard IDA, although this has yet to be explored.

## CONCLUSIONS

In conclusion, we have developed a musculoskeletal modelling and simulation framework for sprinting, which is a highly dynamic locomotor task, using an optimal control theory approach. We quantitatively evaluated the framework's ability to reproduce experimental data, which is a first step that is typically ignored within the sports biomechanics modelling and simulation community. The results from the evaluation suggest that the framework can reproduce experimental data from several sprinting phases with low tracking errors, such that we can proceed with performing predictive simulations to assess technique modifications in relation to performance. This is also the first study to provide dynamically consistent three-dimensional muscle-driven simulations of sprinting across different phases. Overlaid videos of the experimental and simulated kinematics, in addition to the corresponding data files, can be accessed online at https://doi.org/10.6084/m9.figshare.12656354.

## ACKNOWLEDGEMENTS

The authors would like to thank A. Wallbaum, L. Needham, M. Long, O. Okkonen and J. Cowburn for their assistance with data collection.

### Funding

This study was funded by CAMERA, the RCUK Centre for the Analysis of Motion, Entertainment Research and Applications, EP/M023281/1. The funders had no role in study design, data collection and analysis, decision to publish, or preparation of the manuscript.

## Grant Disclosures

The following grant information was disclosed by the authors:
CAMERA.
RCUK Centre for the Analysis of Motion, Entertainment Research and Applications: EP/M023281/1.

## Competing Interests

The authors declare that they have no competing interests.

## Author Contributions

- Nicos Haralabidis conceived and designed the experiments, performed the experiments, analyzed the data, prepared figures and/or tables, authored or reviewed drafts of the paper, and approved the final draft.
- Gil Serrancolí conceived and designed the experiments, analyzed the data, authored or reviewed drafts of the paper, and approved the final draft.
- Steffi Colyer conceived and designed the experiments, performed the experiments, analyzed the data, authored or reviewed drafts of the paper, and approved the final draft.
- Ian Bezodis conceived and designed the experiments, performed the experiments, authored or reviewed drafts of the paper, and approved the final draft.
- Aki Salo conceived and designed the experiments, performed the experiments, authored or reviewed drafts of the paper, and approved the final draft.
- Dario Cazzola conceived and designed the experiments, performed the experiments, analyzed the data, authored or reviewed drafts of the paper, and approved the final draft.

## Human Ethics

The following information was supplied relating to ethical approvals (i.e., approving body and any reference numbers):

Ethical approval for this study was obtained from the University of Bath's Research Ethics Approval Committee for Health (EP 17/18 238).

## Data Availability

Videos of simulated and experimental data, the OpenSim model, and associated files to reproduce the videos are available at Figshare:

Haralabidis, Nicos (2020): Experimental and simulated data, and model corresponding to manuscript. figshare. Media. DOI 10.6084/m9.figshare.12656354.v1

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
