# Peer review of "Three-dimensional data-tracking simulations of sprinting using a direct collocation optimal control approach"

_PeerJ, doi:10.7717/peerj.10975_

## Round 0.1 · original submission · Major Revisions

The reviewers see value in this study, as do I, but have numerous in-depth constructive critiques for improving it, which will require substantial revision to the text, at least; and re-review. We look forward to the revised MS.

·

Basic reporting

No comment.

Experimental design

No comment.

Validity of the findings

The work does not support the conclusion that this model of sprinting is suitable for research on sprint performance. At the very least, the limitations should be discussed. See general comments for details.

Additional comments

This manuscript presents simulations of sprint running, developed using a model of multibody dynamics with contact. The joint torques and contact model parameters were optimized to track kinematic data, measured ground reaction forces, and measured joint torques. It was found that the simulations could closely reproduce the data, with low dynamic inconsistencies. The authors concluded that the model is ready to be used for predictive simulations to study the cause-effect relationships in sprint performance. The paper was technically sound, well written and a pleasure to read.

I agree with the authors that a data-tracking optimization, as presented here, is a necessary and important test to determine whether a model has the capability to replicate a measured performance. Until a model passes this test, applications and predictive simulations should not be considered. However, I feel that this capability was not sufficiently tested, see comments 1 and 2, below.

1. The model is actuated by joint torques rather than muscles. Muscle-actuated simulation is already state of the art, for example, in the work by Falisse which is cited in the manuscript, and which shares one co-author with this manuscript. Muscle actuation would constrain the joint torques to the dynamic force-generating capabilities of muscles. Without constraints on the torques, almost any movement is possible, making the findings hardly surprising. This also is important for the intended application. If one wanted to study sprint performance, performance is only limited by arbitrary bounds placed on the joint torques, rather than bounds informed by muscle physiology. It is unlikely that such a model would provide insight into the factors that affect sprint performance. This limitation diminishes my enthusiasm for this work, and should, at very least, be properly discussed.

2. In the Introduction, the authors mention correctly that dynamic residuals should not be neglected in sports biomechanics research. However, their model still allows non-zero residuals, although they are minimized during the optimization process. If residuals are allowed, the movement is no longer strictly constrained by the laws of physics. Trajectory optimization with zero residuals is the state of the art, (e.g. the cited papers by Falisse) so there is hardly a good excuse for allowing residuals. This is also important for the envisioned applications. If a predictive simulation allows an external “helping hand” force acting on the pelvis, this becomes a major confounding factor for performance (running speed). The residuals that were found to be acceptable in this paper (34 Nm RMS in anterior-posterior direction, Table 3) are as large as the air drag force during sprinting, which is known to be a major factor in performance. I suspect that removing the external pelvis actuation would still produce very good results, and I strongly recommend making that change.

3. The description of the model did not mention air drag, so I assume it was not included. Air drag produces a significant amount of force during sprinting. A simple lumped force, proportional to squared velocity, could be modeled, for instance, based on Quinn, Journal of Sports Science, 2004. The lack of air drag in the model may well be the reason for the large anterior-posterior force residuals (Table 3). This is also important for the intended application: without air drag, performance will be overestimated. At the very least, this limitation should be properly discussed.

4. Line 201-202, “The generic modified model was linearly scaled to match the anthropometric and inertial characteristics of the athlete by using a measurement-based approach within OpenSim”. This requires a citation if it was a specific, previously described approach. If not, a more detailed description is needed.

5. Line 259. The optimization objective has five terms. The first three are data tracking terms, and it is logical to give them the same weight, with exception of the justified lower weight of the torque tracking term. The 4th term penalizes residuals, and it is not automatically clear that its weight (w1) should be the same as the angle and GRF tracking terms. I would prefer not to have this term at all (see comment 2), but if you have it, its weight should not be tied to the tracking term. I suggest defining w1 for angle and GRF tracking, w2 for torque tracking, w3 for residuals, and w4 for accelerations. Then define w = [0.1 0.01 0.1 0.0001]. This does not affect the results, but would prevent the reader from thinking that residuals need to be weighted the same as tracking data.

6. Line 273. If all variables were normalized to 10% of the range observed in the experiment, a constant factor 0.1 should be added in each denominator in the equation. This seems arbitrary. However, this would simply scale the whole objective function by a factor 100, and not affect the optimization problem in any way. Consider leaving out this mention of the 10% because it does not matter. If the scale factor of 100 was helpful to make IPOPT perform better, it can simply be mentioned as an overall scale factor, rather than applied to the denominator in each term. But be careful to account properly for the 5 mm normalization in the anterior-posterior translation variable.

7. Line 294-295. The optimization of the contact model parameters is a strong and innovative aspect of this work. The resulting contact model makes sense, and is probably going to improve the validity of predictive simulations. Consider highlighting this a bit more as one of the goals of the work.

·

Basic reporting

Suggest avoiding use of “excellent” as a subjective descriptor of the quality of the results (Lines 50, 312, 414, 423, 451) or similarly “great” (Line 443). Some context should be provided to the reader to gauge this, e.g. are the errors small vs. measurement error, or vs. trial-to-trial variation, or something similar?

Experimental design

No comment

Validity of the findings

No comment

Additional comments

The study used a direct collocation approach to perform data-tracking simulations of sprinting. It demonstrates a nice alternative to performing inverse dynamics in cases where dynamics residuals are large, among other potential untested use-cases e.g. predictive simulations.

I had five concerns of a technical nature described below that I would like to see addressed in revisions to the manuscript text, and a few other minor suggestions:

(1) Lack of muscles: The planned next step in the research line (as per Lines 57, 113, 131, 147, 443, 558) is predictive simulations of sprinting, which makes the lack of muscles in the present model an important consideration (I'll refrain from citing a bunch of studies on how muscles are important for sprinting). For a non-specific use-case, it's unlikely that the present model lacking muscles would produce realistic results in predictive simulations (it would be interesting if this was the case, but this would likely also require comparison to results with muscles). Essentially what is shown is an alternative approach (vs. traditional inverse dynamics) for estimating joint moments from measured kinematics and GRF, with the benefit of lacking residuals (see question #2 below however). This alone is a fine contribution, I personally would really like to see this become a field-standard approach for doing inverse dynamics, but lacking muscles I have a hard time seeing how the present work demonstrates readiness for predictive simulations. For that I think it would need to be shown here that there is good tracking of the data well with muscles or something like muscles, realistic muscle excitations etc. So I suppose my specific suggestion/question here is, should the paper be cast as an “alternative to inverse dynamics” paper, and not a “ready for predictive simulations” paper (assuming the authors are uninterested in a revision in adding muscles, which would be a major effort)?

(2) Residuals confusion: I was confused by the inclusion of “residual forces” on the pelvis in the model. This may just be a semantics issue and I'm misunderstanding what the authors mean by "residuals", or maybe I’m ill-informed on when “residuals” (F – ma = non-zero) are present in simulations like this, but regardless I though this issue needs to be clarified. In the typical OpenSim workflow or traditional Bresler & Frankel / Winter-style inverse dynamics of gait, residual forces result from applying measured GRF to the model, which were not generated by the model’s dynamics and are not necessarily consistent with its motion, resulting in violations of the equations of motion (Hatze’s (2002) “fundamental problem”). In a typical direct collocation problem, these violations are eliminated by enforcing the equations of motion as constraints. This was done here (Line 286) so it was unclear to me why any additional forces were applied to the pelvis and minimized in the cost function. It seemed that the “residuals” referred to in the simulations were actually forces applied to the pelvis that help move the model independent of the joint moments (sometimes called "hand of god" forces in modeling), not violations of the equations motion that were computed from an inverse dynamics approach. More explanation is needed on what these residual forces on the pelvis actually were and how/if they are consistent with the Hatze/OpenSim definition of residuals. Essentially, I did not understand how a simulation with F = ma as a constraint can have violations of F = ma, which seemed to be the definition of “residuals” in the paper (Line 99).

(3) Optimizing foot-ground contact locations: This approach seemed to deviate from what is generally good modeling practice (assigning model parameters using an independent set of data from the test data). The optimized element locations and stiffness/damping values should be validated in some way, i.e. were the optimized locations realistic locations of contact, were the stiffness/damping realistic for a human plantar foot/shoe/ground composite? Aerts & De Clercq (1993) is a common reference for the latter. The description of the contact model mechanics (Lines 245-253) would benefit from being expanded, e.g. including the mechanics equations and the range of permissible values in the optimization. The shear contact/friction forces and the associated parameters should be included. The finding that different locations are determined from optimizing all phases vs. a single phase is important, but at a more basic level I think the motivation/benefit of optimizing the contact element positions vs. the typical approach just assigning the positions ahead of time to reasonable anatomical locations and avoiding the possibility that the optimizer places them in an unrealistic location, needs to be strengthened. I am not saying "Don't optimize the contact locations", just that I don't see the benefit/need of doing this unless there are too few elements for realistic general tracking, and am wondering if readers will not see it either.

(4) Mesh sensitivity: The choice of the temporal mesh interval density (Line 282) should be justified, e.g. is 80 intervals large enough for “mesh-independent” solutions? This is important to demonstrate before moving onto predictive work. If I understood correctly, the solution was evaluated at ~4*80 nodes, which is likely plenty for a partial gait cycle even of a fast movement like sprinting, but some verification that this is so should ideally be done.

(5) Initial guess: I couldn’t find any mention of the source of the initial guess to the optimizations. This is an important detail in these types of problems.

Minor:
(1) Readers may not understand why it's helpful to have joint torque controls (e.g. why not just compute torques from the state?) and similarly GRF controls. Consider adding an explanation for why this is helpful.

(2) Some additional discussion on this approach vs. other residual reduction/elimination approaches would be helpful, e.g. pros/cons vs. OpenSim RRA.

(3) On the convergence tolerance (Line 293), some additional context on this value would be helpful for readers who are unfamiliar with these terms and methods, e.g. does this mean the largest constraint violation could be 0.001 N?

References
Aerts (1993): https://pubmed.ncbi.nlm.nih.gov/8301705/
Hatze (2002): https://pubmed.ncbi.nlm.nih.gov/11747889/

---

## Round 0.2 · Minor Revisions

The reviewers agree that revisions have been very well done. 1 reviewer has a major concern though about how Fo and Vmax were assumed to be too high, which needs dealing with somehow in revisions. Further review may not be necessary if a convincing solution can be found. We look forward to the revised MS.

·

Basic reporting

No comment.

Experimental design

No comment.

Validity of the findings

No comment.

Additional comments

The authors have responded exceptionally well to the initial review. I have no further comments on the manuscript.

·

Basic reporting

no comment

Experimental design

no comment

Validity of the findings

no comment

Additional comments

This was a well-done revision. The inclusion of muscles concurrent with the multibody dynamics and air drag are very nice additions to the model. Two remaining concerns from this reviewer, the first minor and the second less minor:

(1) Line 353 states “The first set [of equality constraints] enforced that the pelvis residuals obtained from evaluating the skeletal dynamics equations were zero.”

If I understood correctly, this constraint refers to the first six of the 37 multibody dynamics equations, relating to the pelvis DoF where no generalized forces act. I think it may be confusing to readers to describe this constraint separately from the other 31 multibody dynamics constraints, assuming I understood the method correctly and the “pelvis residuals” constraints are indeed these six multibody dynamics equations. Rather than describing this constraint as what it results in (pelvis residual elimination), I suggest instead or also describing its form along with the other multibody dynamics constraints, and presenting the full set of constraints in equation form for clarity. Van den Bogert et al. (2011) is a good example, or Ackermann & van den Bogert (2009) although the latter is an explicit formulation. I may have missed it but I did not see it stated directly in the text that the full multibody dynamics were implemented as constraints.

(2) I apologize for the length of this second comment. I tried to make it as short as possible but ended up needing a lot of text to make and support my point.

Even recognizing that the athlete studied here is elite (10.33 sec 100-m), the strength and power capabilities endowed to the model’s muscles through the maximum isometric force (Fo) and maximum fiber velocity (Vmax) parameters did not seem realistic. I think these choices need greater justification and consideration than what was given beyond referencing the Lai and Dorn studies which used different methods (more on that below).

I will give an example demonstrating why I think the model’s muscle strength is not realistic, using the quadriceps muscles. Handsfield et al. (2014, 2017) presents muscle volume data for the lower limb in a group of healthy young adult “controls” and in competitive sprinters similar in class to the athlete studied here (NCAA Division I). Assuming a typical muscle density of 1056 kg/m^3, the unilateral quadriceps mass was 2.7% of the body mass for controls and 3.6% for sprinters. Assuming a typical specific tension of 300,000 N/m^2, the quadriceps mass in Hamner’s model is 2.5% of the body mass, similar to Handsfield’s controls. Tripling Hamner’s Fo’s by tripling muscle mass produces a quadriceps mass of 7.4% body mass. Even though the Handsfield sprinters were likely all from the same college team and likely did not all have ~10.3-sec speed, it does not seem likely that the athlete here had double their quadriceps mass relative to body mass. Tripling Hamner’s Fo’s while matching the 3.6% from Handsfield requires a specific tension of about 620,000 N/m^2. This specific tension is not outside the realm of possibility and similar values have been used in other models (e.g. Rajagopal et al., 2016), but regardless of how the tripling of Fo is achieved, it produces a model with unbelievable strength. The maximum isometric knee extension torque of Hamner’s model with tripled Fo’s is 718 Nm or about 10 Nm/kg. Elite sprinters in dynamometry studies typically produce only about 200-300 Nm and it is rare to see any athlete produce a max isometric torque much above 400 Nm. For example, even assuming a very long moment arm of 0.5 m, the youngest (age 43) elite masters Olympic weightlifters in Pearson et al. (2002) averaged 618 N x 0.5 m = 309 Nm of torque or 3.5 Nm/kg. Even considering things like antagonism and lack of true maximal activation in human data, 700+ Nm is an unbelievable level of strength.

The assumed value of Vmax = 20 is also not realistic. There are no studies on human muscle to my knowledge reporting Vmax much faster than 10-12. Such an extreme value of Vmax should not be needed to simulate elite speeds. Previous optimal control simulation studies (with much coarser controls) have reached world-class speeds with Vmax = 10-12, without super-human muscle strengths (Sellers & Manning, 2007; Miller et al., 2012).

The values used for Fo and Vmax were justified in the text as being consistent with previous modeling studies on fast running/sprinting (Lai and Dorn references). Dorn indeed used triple the Hamner Fo’s and Vmax = 20, but Dorn was also solving an inverse dynamics problem, where the model is forced to produce joint moments computed from experimental GRF that result in the Hatze “Fundamental Problem” in my first review, leading to the need for things like reserve actuators and unrealistic muscle properties to compute muscle activations less than 1.0. Lai was also solving an inverse dynamics problem and also used larger values of Fo and Vmax than Hamner but should not be cited to support the values used in the present study. Lai did not use triple Hamner’s Fo values, and used Vmax = 15 not 20. The issues in Lai and Dorn’s methods that required these things do not affect the present methods, the model can deviate from the measured joint moments a little and still track them reasonably well overall. Relatedly, the need for reserve actuators and extreme muscle parameters in the standard OpenSim IK/RRA/CMC workflow typically appears in low-force situations when the muscle model is in unusual states, not in high-force situations which is where an optimal control simulation would likely benefit from these things.

I mention all of these things (again apologizing for the length of this comment/rant) not because I think the present work is heavily flawed (it’s great work), but because there seemed to be several elements in the present model that were included not for physiological reasons but for fear that the model will not track the data well: reserve actuators, unrealistic Fo’s, unrealistic Vmax. These things should not be needed here for this model to produce quality tracking simulations. If the authors have not already done so, I would encourage them to check if the model can track the data reasonably well with more reasonable values of Fmax (e.g. maybe 1.5-2.0 Hamner’s values, or the values from Rajagopal where max knee torque is a less extreme 470 Nm), Vmax = 12, and no reserve actuators, and if this is possible, replace the present results with those results. If the model is not able to track the data well without 3xFo and Vmax = 20, this is concerning and the reason for this should be investigated, as these values will likely produce inhumanly fast sprinting speeds in predictive simulations. It would be helpful to include some justification of the parameter values, e.g. in reference to the Handsfield et al. (2017) data, Vmax studies, and studies like Erskine et al. (2011) on specific tension increasing with strength training, rather than references to Lai and Dorn which are great studies are not very relevant for justifying parameter values when using direct collocation methods.

---

## Round 0.3 · accepted · Accept

I have checked the revised MS/Response and am convinced that the authors have done a good job attending to the prior critiques. The study is very interesting and a great contribution to PeerJ. Congratulations!!